# DEMYSTIFYING CLIP DATA

**Hu Xu**[1] **Saining Xie**[2] **Xiaoqing Ellen Tan**[1] **Po-Yao Huang**[1] **Russell Howes**[1] **Vasu Sharma**[1]
**Shang-Wen Li**[1]      **Gargi Ghosh**[1]      **Luke Zettlemoyer**[1,3]      **Christoph Feichtenhofer**[1]
[1]FAIR, Meta AI      [2]New York University      [3]University of Washington

## ABSTRACT

Contrastive Language-Image Pre-training (CLIP) is an approach that has advanced research and applications in computer vision, fueling modern recognition systems and generative models. We believe that the main ingredient to the success of CLIP is its *data* and *not* the *model* architecture or pre-training objective. However, CLIP only provides very limited information about its data and how it has been collected, leading to works that aim to reproduce CLIP's data by filtering with its model parameters. In this work, we intend to reveal CLIP's data curation approach and in our pursuit of making it open to the community introduce Metadata-Curated Language-Image Pre-training (MetaCLIP). MetaCLIP takes a raw data pool and metadata (derived from CLIP's concepts) and yields a balanced subset over the metadata distribution. Our experimental study rigorously isolates the model and training settings, concentrating solely on data. MetaCLIP applied to CommonCrawl with 400M image-text data pairs outperforms CLIP's data on multiple standard benchmarks. In zero-shot ImageNet classification, MetaCLIP achieves 70.8% accuracy, surpassing CLIP's 68.3% on ViT-B models. Scaling to 1B data, while maintaining the same training budget, attains 72.4%. Our observations hold across various model sizes, exemplified by ViT-bigG producing 82.1%. Curation code and training data distribution over metadata is available at https://github.com/facebookresearch/MetaCLIP.

## 1 INTRODUCTION

Deep learning has revolutionized the field of artificial intelligence, and pre-trained models have played a pivotal role in democratizing access to cutting-edge AI capabilities. However, the training data used to create these models is often concealed from the public eye, shrouded in secrecy.

The increasing availability of pre-trained models for public use contrasts sharply with the lack of transparency regarding their training data. Further, proprietary concerns, such as copyright issues, often limit access to the original data sources. Consequently, the need to explore novel approaches for curating high-quality training data that can be shared openly arises.

In the vision-language domain, the dominant model and learning approach is Contrastive Language-Image Pre-training (CLIP) (Radford et al., 2021), a simple technique to learn from image-text pairs. We believe that the secret to the dominance of CLIP models is attributed to its high-quality WIT400M *dataset* which is curated from the web. Despite its popularity, the specifics of CLIP's curation process have remained a mystery, captivating the research community for years.

Follow-up works (Schuhmann et al., 2022; 2021) have attempted to replicate CLIP's data, but with a notable difference in their curation method. While CLIP generates data based on its unknown data source and curation methodology, these approaches remove noise by applying the CLIP model as a hard blackbox filter which in turn is a form of distilling WIT400M information captured in CLIP.

The advantages of CLIP's curation are apparent. First, it starts *from scratch*, avoiding the introduction of biases through filters. Second, CLIP's curation process *balances* the data distribution over metadata, maximizing signal preservation while mitigating, rather than removing, noise in the data[1]. Such distribution lays the groundwork for task-agnostic data, a crucial part of foundation models.

---

[1]For example, a filter on digits can remove noise from date or id strings but remove signal for tasks that involve OCR (e.g., MNIST), or a filter removing text with less than 5 characters can remove signal "dog".

In this paper, we attempt to *reveal* CLIP's method around training *data curation*. We present an empirical study on data curation, with frozen model architecture and training schedule. We focus solely on the impact of training *data*, excluding other factors that could confound the results. We make several observations for good data quality and present a simple algorithm to make CLIP's curation more transparent. Consequently, we shed light on both the curation process and the resulting training data *distribution*. Our algorithm enables easy adaptation to different data pools, allowing parties to fully own their data pipeline without relying on blackbox filters from external providers.

Our algorithm takes a raw data pool $\mathcal{D}$ and metadata $\mathcal{M}$ (derived from CLIP's queries or visual concepts) and yields a balanced subset $\mathcal{D}^*$ over $\mathcal{M}$: $\mathcal{D}^* \leftarrow f(\mathcal{D}; \mathcal{M})$. Our approach, named Metadata-Curated Language-Image Pretraining (MetaCLIP), marks a significant step towards making the curation process more transparent and accessible.

MetaCLIP applied to CommonCrawl (CC) with 400M data points outperforms CLIP on multiple standard benchmarks. In terms of zero-shot ImageNet classification, using ViT (Dosovitskiy et al., 2020) models of various sizes. Our MetaCLIP

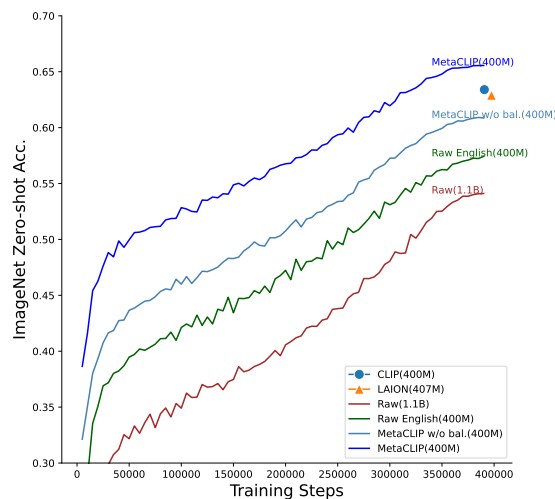

Figure 1: ViT-B/32 on ImageNet zero-shot classification with fixed training steps (12.8B seen pairs and training/validation data has been de-duplicated). Raw: raw CommonCrawl (CC) distribution; Raw English: English only CC; MetaCLIP w/o bal.: curated (substring matched) data pool from CC; MetaCLIP: curated *and balanced* metadata distribution. Metadata curation boosts performance significantly and balancing is equally important. Our MetaCLIP data significantly outperforms CLIP's WIT400M and LAION data(Schuhmann et al., 2021).

achieves 70.8% vs CLIP's 68.3% on ViT-B and 76.2% vs 75.5% on ViT-L. Scaling to 2.5B data, with the *same* training budget and similar distribution boosts this to unprecedented accuracy of 79.2% for ViT-L , 80.5% for ViT-H and 82.1% for ViT-bigG in the vanilla training setting (not using any external data, models, or longer training).

In Fig.1, we show the impact of metadata curation on ImageNet validation plotted over training steps. First, we are training on Raw English data from the web (400M image-text pairs, 57.4% accuracy), after applying Language IDentification (LID) to the random Raw set (∼1.1B pairs, 54.1%). Using metadata to curate the training set (MetaCLIP 400M w/o bal, 60.8%) performs significantly better than these baselines, and using balancing significantly increases accuracy further (MetaCLIP, 65.5%), outperforming similar datasets, WIT400M from CLIP, 63.4% and LAION 400M, 60.0%(Schuhmann et al., 2021).

## 2    RELATED WORK

The training data of CLIP differs significantly from a traditional supervised dataset (Gadre et al., 2023) in various aspects. Firstly, it involves large-scale training with mixed-quality image-text pairs rather than categorized images with human annotated labels, as commonly seen in classification datasets. Secondly, CLIP's pre-training is the initial stage of training, assuming no access to previously trained models.

**Data Pruning on Established Datasets.** Current research on data algorithms primarily revolves around *data pruning* techniques applied to well-established datasets using pre-trained models (Sorscher et al., 2022; Abbas et al., 2023). These approaches, such as coreset selection techniques (Har-Peled & Mazumdar, 2004; Feldman et al., 2011; Bachem et al., 2015; Mirzasoleiman et al.,

2020; Toneva et al., 2018), aim to select a subset of data that yields similar performance to training on the entire dataset. Post-hoc data pruning with model filters has limited utility, if the model is used as a black-box filter that forbids to control biases or improve the filter quality.

**Handling Noisy Internet Data.**    Addressing noisy data from the Internet is a significant challenge, and existing approaches often heavily rely on human-designed filter systems. Classical methods involve dataset cleaning and outlier removal (Jiang et al., 2001; Yu et al., 2002) to discard samples that may introduce undesirable biases to models.

**Replicating CLIP's Training Data.**    Recent efforts, such as LAION (Schuhmann et al., 2021; 2022) and concurrent work DataComp (Gadre et al., 2023), attempt to replicate CLIP's training data. However, they adopt fundamentally different strategies for several reasons. First, the data used in these approaches are post-hoc, filtered, by vanilla CLIP as a *teacher* model. Second, the curation process in these methods relies on a labor-intensive pipeline of filters, making it challenging to comprehend the resulting data distribution from the raw Internet (refer to the unknown biases of using CLIP filter in (Schuhmann et al., 2022)). Thirdly, the goal is to match the quantity of CLIP's target data size rather than the data distribution itself, which may lead to an underestimation of the data pool size needed to obtain sufficient quality data. Consequently, the performance on the 400M scale is sub-optimal, with LAION400M only achieving 72.77% (Schuhmann et al., 2021) accuracy on ViT-L/14 on ImageNet, whereas vanilla CLIP obtains 75.5%.

**Importance of Understanding CLIP's Data Curation.**    The observations made in these studies underscore the critical importance of understanding how OpenAI CLIP curates its data in the first place. A comprehensive understanding of the curation process can shed light on the factors that contribute to its success, allowing researchers to devise more effective and efficient algorithms for future vision-language pre-training endeavors.

## 3    METACLIP

The original paper (Radford et al., 2021) only provides limited details about how CLIP curates its data. Since important design choices for a direct reproduction are missing, we will clarify our choices in this section. Our goal is to uncover CLIP's data curation process, which involves preserving signal in the data while minimizing noise. In this section, we will explain the principles we have adopted to achieve this, which may differ from CLIP's as these are not known publicly.

CLIP's WIT400M is curated with an information retrieval method, quoting (Radford et al., 2021):

> "
>     To address this, we constructed a new dataset of 400 million (image, text) pairs collected from a variety of publicly available sources on the Internet. To attempt to cover as broad a set of visual concepts as possible, we ***search*** for (image, text) pairs as part of the construction process whose text includes one of a set of *500,000 **queries*** We approximately class balance the results by including *up to 20,000 (image, text) pairs per query*.
> "

We rigorously adhere to this description and provide detailed insights into the construction process of CLIP's metadata (in §3.1)[2], sub-string matching (in §3.2), inverted indexing (in §3.3), as well as query and balancing (in §3.4).

### 3.1    METADATA CONSTRUCTION: $\mathcal{M} = \{entry\}$

We start by re-building CLIP's 500,000-query metadata, citing Radford et al. (2021):

---

[2]We generalize the term queries (used by CLIP) as *entries* in *metadata* because metadata describe training data and our algorithm does not require search on inverted index yet have similar effects.

> "
> The base query list is all words occurring at least 100 times in the *English version of Wikipedia*. This is augmented with *bi-grams* with high pointwise mutual information as well as the names of all *Wikipedia articles* above a certain search volume. Finally all *WordNet synsets* not already in the query list are added.
> "

The metadata ('queries' or 'entries') consists of four components: (1) all synsets of WordNet, (2) uni-grams from the English version of Wikipedia occurring at least 100 times, (3) bi-grams with high pointwise mutual information, and (4) titles of Wikipedia articles above a certain search volume. We rebuild these components from WordNet and Wikipedia and summarize the statistics in Table 1[3]. We estimate the thresholds for components (3) and (4) as in the 3rd column of Table 1, by first choosing a point-wise mutual information threshold of 30 that meets the budget of 100k entries for bi-grams and then fill the rest of the entries with Wikipedia titles.

| Source | # of Entries | Desc. of Threshold | Threshold |
|---|---|---|---|
| WordNet synsets | 86,654 | N/A | [ALL] (follow CLIP) |
| Wiki uni-gram | 251,465 | Count | 100 (follow CLIP) |
| Wiki bi-gram | 100,646 | Pointwise Mutual Info.(PMI) | 30 (estimated) |
| Wiki titles | 61,235 | View Frequency | 70 (estimated) |

Table 1: Composition of MetaCLIP Metadata.

## 3.2 SUB-STRING MATCHING: *text → entry*

After constructing the metadata, CLIP's curation aligns a pool of image-text pairs with metadata entries through sub-string matching. This process identifies texts that contain any of the metadata entries, effectively associating unstructured texts with structured metadata entries. The sub-string matching step retains only high-quality matching texts, automatically filtering out various types of noises that a typical filter system would consider on a case-by-case basis.

Such alignment is referred to as sub-string matching in Radford et al. (2021):

> "
> We also restrict this step in CLIP to text-only querying for *sub-string matches* while most webly supervised work uses standard image search engines ...
> "

**Image-Text Pair Pool** We start by estimating the pool size used by CLIP's curation. CLIP's data source is unknown to us ("a variety of publicly available sources" in Radford et al. (2021)). We adopt CommonCrawl (CC)[4] as the source to build such a pool and re-apply sub-string matching to this source. We ended with a pool of 1.6B image-text pairs (5.6B counts of sub-string matches). Note that one text can have multiple matches of entries and we have 3.5 matches per text on average.

As a result, sub-string matching builds the mapping *txt → entry*. This step has two outcomes: (1) low-quality text is dropped; (2) unstructured text now has a structured association with metadata. For all English text, ∼50% image-text pairs are kept in this stage. Similar to CiT (Xu et al., 2023), this approach looks for quality matches and automatically gets rid of some type of noise (such as date strings) that a typical filter system would remove consider case-by-case (e.g., regular expression on dates, ids etc.).

---

[3]Note that we cannot find Wikipedia's search volume for titles of Wikipedia (4). Instead, we use volumes of Pageviews on Wiki articles. We randomly selected 26 days' Pageviews from Apr. 2018 to Sep. 2022.

[4]https://commoncrawl.org

| Metadata Subset | # of Entries | # of Counts |
|---|---|---|
| Full | 500K | 5.6B |
| Counts = 0 | 114K | 0 |
| Counts > 20000 | 16K | 5.35B |

Table 2: Summary of counts for entries.

| Entry | Counts | Entry | Counts | Entry | Counts | Entry | Counts |
|---|---|---|---|---|---|---|---|
| of | 120M | in | 107M | and | 100M | for | 89M |
| the | 87M | The | 67M | with | 67M | to | 61M |
| photo | 54M | a | 50M | image | 48M | 1 | 47M |
| on | 45M | by | 43M | 2 | 43M | Image | 39M |
| at | 38M | Black | 33M | 3 | 30M | A | 29M |

Table 3: Top-20 entries with counts.

## 3.3 INVERTED INDEXING: *entry → text*

Following sub-string matching, CLIP builds an inverted index of the data pool. All texts associated with each metadata entry are aggregated into lists, creating a mapping from each entry to the corresponding texts, *entry → text*.

As an analysis, we count the number of matches for each entry and summarize that in Table 2. The counts exhibit a long-tailed distribution. Out of the 500k entries, **114k** entries have *no* matches. This signifies the importance of knowing the training data distribution since it is very likely the training data does not have certain visual concepts. We observed that only 16k entries had counts higher than 20k, accounting for only **3.2%** (16k/500k) of the entries, but their counts made up **94.5%** (5.35B/5.6B) of the total counts of all entries.

**Top Entries.** We show the top entries of the matching in Table 3. Interestingly, many of these are stopwords, which don't carry specific meaning but can enhance the overall text quality (e.g., by generating grammatically correct sentences rather than just keyword lists). It's important to note that although sub-string matching aims to select only high-quality texts, there are instances where common entries may still include irrelevant texts. For instance, the entry "photo" could match with the popular but unhelpful term "untitled photo". These noise-related issues can be addressed in the subsequent stage of processing.

## 3.4 QUERY AND BALANCING WITH $t \leq 20k$

The key secret behind OpenAI CLIP's curation is to balance the counts of matched entries. For each metadata entry, the associated list of texts (or image-text pairs) is sub-sampled, ensuring that the resulting data distribution is more balanced. This step aims to mitigate noise and diversify the distribution of data points, making the data more task-agnostic as foundation data for pre-training.

The magic number $t = 20k$ is a threshold used to limit the number of texts/pairs for each entry. Entries with fewer than $t$ pairs (tail entries) retain all associated pairs, while entries with more than $t$ pairs (head entries) are sub-sampled to $t$ pairs. The selection is based on the density of information in texts; texts with more matched entries have a higher chance of being curated (recall that the average is 3.5 matches per text).

To study the effect of the magic number $t = 20k$, we plot the cumulative sum of counts for entries sorted by counts from tail to head in Fig. 2. Interestingly, the value of $t = 20k$ seemingly represents the transition from tail to head entries, when the head entries start exhibiting an *exponential growth rate*. By applying a max count of $t$, the growth rate of total counts (i.e., the scale of resulting data points) is reduced to *linear*. This significantly flattens (and balances) the training data distribution. We further study the optimality of $t = 20k$ for the 400M data scale in our experiments.

In summary, balancing yields three interesting outcomes:

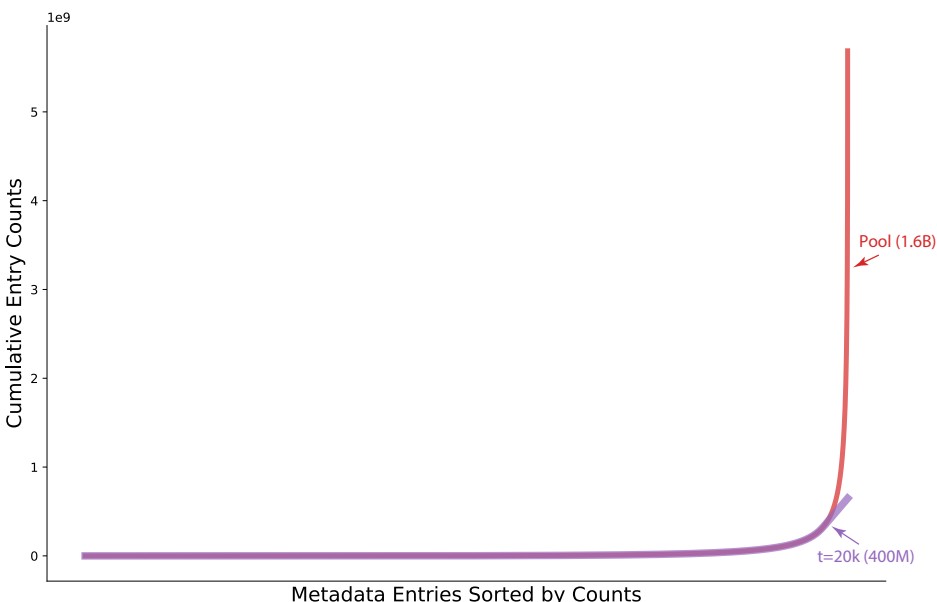

Figure 2: Cumulative sum of counts on entries from *tail to head* on a data pool with 1.6B image-text pairs (5.6B match counts). (1) raw/unbalanced cumulative counts, $t = \infty$; (2) balanced cumulative counts after applying $t = 20$k. The limit $t$ defines the transition of tail/head entries.

(i) It reduces dominance and noise from head entries, like common web terms. E.g., out of 400M pairs, only 20k texts containing "photo" are kept (while there are 54M "photo" instances in the pool).

(ii) It diversifies the data distribution and balances tail/head entries, leading to a more task-agnostic foundation.

(iii) Sampling for each entry ensures that data points with more matched entries or *denser* information are prioritized for curation.

**Discussion.** CLIP employs a pure NLP-based approach, requiring no access to ML models and minimizing explicit/implicit priors from humans. The metadata plays a central role in mitigating noise and preserving signal in the data distribution. The balancing step effectively flattens the data distribution, diversifying the data and making it more suitable as foundation data for pre-training tasks. We analyze the effects of balancing in Appendix A.3.

### 3.5   A SIMPLE ALGORITHM FOR CURATION

This section presents an algorithm that formalizes the curation process described earlier. The algorithm aims to improve scalability and reduce space complexity for operations across data points, such as inverted indexing and sub-sampling. Instead of building inverted indexes, the algorithm only maintains total counts for each entry.

We assume that CLIP curation constructs an inverted index that maps entries to documents (image-text pairs) to enable efficient *search* for each entry ("we search for (image-text) pairs" in Radford et al. (2021)). In contrast, our algorithm approaches the balancing process through independent sampling. This avoids the need to build an inverted index that could potentially store hundreds of millions of concrete pairs for popular entries, thereby improving efficiency and scalability.

Our algorithm takes three inputs: metadata $\mathcal{M}$, a data pool $\mathcal{D}$, and a hyper-parameter $t$. It aims to find a subset $\mathcal{D}^*$ with a balanced distribution over $\mathcal{M}$, denoted as $\mathcal{D}^* \leftarrow f(\mathcal{D}; \mathcal{M}, t)$. The algorithm consists of two parts, each corresponding to a specific stage of the curation process.

We provide the Python pseudo-code in Algorithm 1.

**Algorithm 1:** Pseudo-code of Curation Algorithm in Python style (see Sec. A.10 for samples).

```python
# D: raw image-text pairs;
# M: metadata;
# t: max matches per entry in metadata;
# D_star: curated image-text pairs;

D_star = []
# Part 1: sub-string matching: store entry indexes in text.matched_entry_ids and
    output counts per entry in entry_count.
entry_count = substr_matching(D, M)
# Part 2: balancing via indepenent sampling
entry_count[entry_count < t] = t
entry_prob = t / entry_count
for image, text in D:
    for entry_id in text.matched_entry_ids:
        if random.random() < entry_prob[entry_id]:
            D_star.append((image, text))
            break
```

**Part 1: Entry Counts from Sub-string Matching.** This corresponds to Sec. 3.2. The `substr_matching` function outputs the total counts of matches per entry, `entry_count`, represented as a NumPy array indexed by `entry_id`. Each `text` is associated with `matched_entry_ids` that contains a list of matched entries.

**Part 2: Balancing via Independent Sampling.** This part corresponds to Sec.3.3 and Sec.3.4 and focuses on balancing counts on entries. Instead of building an expensive inverted index with associated lists of texts for each entry, we sample each data point independently.

We first compute the probability of sampling each entry, `entry_prob`, where tail entries (`entry_count < t`) have a probability equal to 1, and head entries have a probability less than 1. We iterate through all image-text pairs and sample/curate each pair. When an image-text pair has a matched entry sampled/selected, we include that pair in $\mathcal{D}^*$.

This procedure is equivalent to CLIP's curation, because if one image-text pair has one or more matched entries, the chance of that pair being selected is determined by the probability of sampling for each individual entry: $t/\texttt{entry\_count[entry\_id]}$. As long as one entry selects that pair, it will be kept in $\mathcal{D}^*$. Our independent sampling approach allows us to scale balancing for each data point independently and reduces the global operation to counting the total matches for each entry. We demonstrate case studies in experiments on (1) scaling curation in a data pipeline and (2) online balancing in data loader.

## 4 EXPERIMENTS

**Data Pools.** We collect two pools of data:

Pool 1 contains 1.6 billion image-text pairs with a total of 5.6 billion counts of matches. This pool was used to estimate a target of **400M** image-text pairs, collected from 15 snapshots of Common-Crawl (CC) from January 2021 to January 2023.

Pool 2 aims to scale curation in our data pipeline. We parsed all 90 CC snapshots from 2013 to April 2023, using our algorithm (see §A.2 for details on the curation pipeline) to curate from a pool of 10.7B matched image-text pairs that are originally from a large set of URL-text pairs, which have undergone de-duplication, English Language IDentification (LID) and sub-string matching. However, we only perform (expensive) image downloading, storing, and transferring for data points that are distribution-calibrated and selected by our algorithm.

For balancing we consider 2 scenarios on this data: (i) $t = 170k$, which is resulting in **2.5B** image-text pairs. This $t = 170k$ configuration has tail counts amounting to 6% of the total counts, the *same tail/head ratio* that the 400M Pool 1 data has, produced by applying $t = 20k$ on the 1.6B Pool 1 data. (ii) The $t = 20k$ threshold applied to Pool 2 which results in **1B** image-text pairs and compared to the 400M set from Pool 1 only increases tail metadata matches (head counts are capped at $20k$).

| | Average | ImageNet | Food-101 | CIFAR10 | CIFAR100 | CUB | SUN397 | Cars | Aircraft | DTD | Pets | Caltech-101 | Flowers | MNIST | FER-2013 | STL-10 | EuroSAT | RESISC45 | GTSRB | KITTI | Country211 | PCAM | UCF101 | Kinetics700 | CLEVR | HatefulMemes | SST2 |
|---|---|---|---|---|---|---|---|---|---|---|---|---|---|---|---|---|---|---|---|---|---|---|---|---|---|---|---|
| **ViT-B/32** | | | | | | | | | | | | | | | | | | | | | | | | | | | |
| CLIP, our eval. | 56.6 | 63.4 | 83.7 | 89.8 | 65.1 | 53.7 | 62.0 | 59.7 | 19.6 | 44.0 | 87.2 | 87.4 | 66.9 | 48.2 | 46.6 | 97.1 | 44.9 | 61.0 | 32.6 | 28.7 | 17.2 | 62.5 | 63.9 | 48.0 | 23.6 | 56.4 | 58.6 |
| OpenCLIP, our eval. | 57.6 | 62.9 | 80.7 | 90.7 | 70.6 | 61.2 | 66.4 | 79.2 | 16.7 | 54.5 | 86.5 | 90.7 | 66.1 | 37.4 | 48.2 | 95.6 | 52.2 | 58.0 | 42.0 | 38.0 | 14.8 | 50.1 | 63.0 | 42.8 | 22.5 | 53.3 | 52.3 |
| **MetaCLIP** | **58.2** | 65.5 | 80.6 | 91.3 | 70.2 | 63.4 | 63.0 | 70.7 | 26.8 | 52.8 | 88.7 | 91.9 | 68.5 | 41.5 | 35.9 | 95.4 | 52.6 | 64.2 | 35.8 | 30.7 | 17.2 | 55.5 | 66.1 | 45.4 | 30.6 | 56.4 | 53.4 |
| **ViT-B/16** | | | | | | | | | | | | | | | | | | | | | | | | | | | |
| CLIP, our eval. | 59.6 | 68.3 | 88.8 | 90.8 | 68.2 | 55.6 | 64.0 | 64.6 | 24.0 | 45.1 | 88.9 | 89.1 | 69.4 | 51.8 | 53.0 | 98.2 | 54.8 | 65.5 | 43.3 | 21.7 | 22.8 | 56.3 | 68.5 | 52.3 | 25.5 | 58.7 | 60.5 |
| OpenCLIP, our eval. | 60.4 | 67.0 | 85.8 | 91.7 | 71.4 | 65.3 | 69.2 | 83.6 | 17.4 | 51.0 | 89.2 | 90.8 | 66.5 | 56.3 | 46.1 | 97.0 | 52.2 | 65.7 | 43.5 | 23.7 | 18.1 | 51.7 | 60.0 | 46.2 | 33.9 | 54.5 | 54.4 |
| **MetaCLIP** | **61.1** | 70.8 | 86.8 | 90.1 | 66.5 | 70.8 | 66.6 | 74.1 | 27.9 | 55.9 | 90.4 | 93.8 | 72.3 | 47.8 | 44.6 | 97.2 | 55.4 | 68.8 | 43.8 | 33.4 | 22.6 | 52.9 | 68.0 | 49.5 | 22.8 | 54.8 | 60.6 |
| **ViT-L/14** | | | | | | | | | | | | | | | | | | | | | | | | | | | |
| CLIP, our eval. | 65.7 | 75.5 | 93.0 | 95.6 | 78.3 | 63.3 | 66.8 | 77.8 | 31.3 | 55.3 | 93.6 | 93.3 | 79.3 | 76.4 | 56.9 | 99.4 | 61.9 | 70.9 | 50.6 | 19.2 | 31.9 | 50.1 | 75.7 | 60.2 | 22.3 | 59.7 | 68.9 |
| OpenCLIP, our eval. | 64.5 | 72.7 | 90.0 | 94.7 | 78.0 | 73.9 | 72.4 | 89.5 | 24.7 | 60.2 | 91.6 | 93.6 | 73.0 | 76.1 | 54.3 | 98.1 | 63.9 | 69.6 | 49.9 | 16.0 | 23.0 | 51.7 | 71.5 | 51.6 | 25.4 | 55.3 | 56.0 |
| **MetaCLIP** | **67.1** | 76.2 | 90.7 | 95.5 | 77.4 | 75.9 | 70.5 | 84.7 | 40.4 | 62.0 | 93.7 | 94.4 | 76.4 | 61.7 | 46.5 | 99.3 | 59.7 | 71.9 | 47.5 | 29.9 | 30.9 | 70.1 | 75.5 | 57.1 | 35.1 | 56.6 | 65.6 |

Table 4: MetaCLIP-400M *vs.* CLIP (WIT400M data) and OpenCLIP (LAION-400M data(Schuhmann et al., 2021)). We use 3 different model scales (ViT-B/32 and -B/16 and -L/14) and an identical training setup as CLIP.

| | Average | ImageNet | Food-101 | CIFAR10 | CIFAR100 | CUB | SUN397 | Cars | Aircraft | DTD | Pets | Caltech-101 | Flowers | MNIST | FER-2013 | STL-10 | EuroSAT | RESISC45 | GTSRB | KITTI | Country211 | PCAM | UCF101 | Kinetics700 | CLEVR | HatefulMemes | SST2 |
|---|---|---|---|---|---|---|---|---|---|---|---|---|---|---|---|---|---|---|---|---|---|---|---|---|---|---|---|
| **ViT-B/32** | | | | | | | | | | | | | | | | | | | | | | | | | | | |
| MetaCLIP(400M) | 58.2 | 65.5 | 80.6 | 91.3 | 70.2 | 63.4 | 63.0 | 70.7 | 26.8 | 52.8 | 88.7 | 91.9 | 68.5 | 41.5 | 35.9 | 95.4 | 52.6 | 64.2 | 35.8 | 30.7 | 17.2 | 55.5 | 66.1 | 45.4 | 30.6 | 56.4 | 53.4 |
| MetaCLIP(1B) | 60.3 | 67.3 | 81.9 | 95.2 | 76.7 | 71.4 | 65.9 | 73.0 | 31.4 | 58.9 | 89.5 | 92.5 | 72.6 | 35.4 | 45.8 | 96.3 | 50.4 | 64.6 | 40.7 | 32.0 | 17.0 | 64.2 | 70.3 | 47.8 | 14.6 | 54.9 | 56.8 |
| MetaCLIP(2.5B) | 59.8 | 67.6 | 82.6 | 95.2 | 77.7 | 67.8 | 66.8 | 77.2 | 26.9 | 58.9 | 90.9 | 92.5 | 69.7 | 42.7 | 48.3 | 96.3 | 49.9 | 66.5 | 39.2 | 29.3 | 17.7 | 50.0 | 68.0 | 47.6 | 19.4 | 53.5 | 53.1 |
| **ViT-B/16** | | | | | | | | | | | | | | | | | | | | | | | | | | | |
| MetaCLIP(400M) | 61.1 | 70.8 | 86.8 | 90.1 | 66.5 | 70.8 | 66.6 | 74.1 | 27.9 | 55.9 | 90.4 | 93.8 | 72.3 | 47.8 | 44.6 | 97.2 | 55.4 | 68.8 | 43.8 | 33.4 | 22.6 | 52.9 | 68.0 | 49.5 | 22.8 | 54.8 | 60.6 |
| MetaCLIP(1B) | 63.2 | 72.4 | 88.1 | 94.8 | 78.2 | 77.5 | 66.4 | 79.3 | 38.0 | 57.7 | 92.3 | 93.6 | 75.1 | 36.4 | 47.8 | 98.0 | 50.5 | 70.1 | 49.5 | 36.6 | 21.6 | 53.7 | 74.1 | 52.7 | 21.6 | 56.8 | 61.6 |
| MetaCLIP(2.5B) | 63.5 | 72.1 | 88.3 | 95.7 | 79.0 | 71.4 | 68.5 | 82.9 | 30.3 | 62.1 | 91.7 | 93.3 | 73.9 | 66.1 | 47.0 | 98.4 | 51.1 | 71.1 | 46.6 | 16.6 | 22.7 | 50.5 | 73.0 | 52.5 | 30.8 | 57.4 | 59.0 |
| **ViT-L/14** | | | | | | | | | | | | | | | | | | | | | | | | | | | |
| MetaCLIP(400M) | 67.1 | 76.2 | 90.7 | 95.5 | 77.4 | 75.9 | 70.5 | 84.7 | 40.4 | 62.0 | 93.7 | 94.4 | 76.4 | 61.7 | 46.5 | 99.3 | 59.7 | 71.9 | 47.5 | 29.9 | 30.9 | 70.1 | 75.5 | 57.1 | 35.1 | 56.6 | 65.6 |
| MetaCLIP(1B) | 70.2 | 79.0 | 92.9 | 96.8 | 84.9 | 83.1 | 72.8 | 86.5 | 48.9 | 65.9 | 95.3 | 94.8 | 84.7 | 53.8 | 54.1 | 99.3 | 70.0 | 73.8 | 58.7 | 36.3 | 32.2 | 70.4 | 81.4 | 61.6 | 21.1 | 61.2 | 66.1 |
| MetaCLIP(2.5B) | 69.8 | 79.2 | 93.4 | 97.6 | 84.2 | 80.1 | 73.8 | 88.7 | 44.6 | 68.1 | 94.7 | 95.4 | 81.8 | 64.4 | 55.1 | 99.3 | 59.2 | 74.6 | 56.3 | 29.7 | 34.0 | 67.3 | 81.6 | 62.0 | 25.9 | 58.0 | 66.7 |
| **ViT-H/14** | | | | | | | | | | | | | | | | | | | | | | | | | | | |
| MetaCLIP(2.5B) | 72.4 | 80.5 | 94.2 | 98.0 | 86.4 | 83.4 | 74.1 | 90.0 | 50.2 | 72.4 | 95.4 | 95.6 | 85.1 | 72.7 | 55.2 | 99.4 | 66.3 | 74.6 | 62.5 | 38.2 | 37.2 | 65.8 | 82.2 | 64.1 | 30.1 | 59.3 | 69.2 |
| **ViT-bigG/14** | | | | | | | | | | | | | | | | | | | | | | | | | | | |
| MetaCLIP(2.5B) | 73.2 | 82.1 | 94.9 | 98.5 | 88.6 | 84.0 | 74.7 | 90.9 | 52.7 | 72.6 | 96.1 | 95.7 | 89.5 | 78.1 | 56.7 | 99.5 | 73.7 | 75.5 | 61.7 | 31.0 | 41.5 | 65.6 | 85.6 | 65.8 | 24.3 | 58.8 | 65.3 |

Table 5: Scaling MetaCLIP from 400M ($t$=20k) to 1B ($t$=20k) and 2.5B ($t$=170k) training data.

**Training Setup** We strictly follow the CLIP training setup, using V100 32GB GPUs and an equivalent global batch size of 32,768. For ViT-B/32 and ViT-B/16, we use 64 GPUs with a per GPU batch size of 512 and for ViT-L/14 we use 128 GPUs with a 256 per GPU batch size. It takes 4 days to train ViT-B/32 and a month to train ViT-L/14. We use 256 A100 80GB GPUs to train ViT-H/14 and ViT-bigG/14 model for 1 week and 2 months, respectively. We train in all experiments for the same number of iterations that correspond to 12.8B seen image-text pairs during training (32 epochs for 400M). We pre-process with face-blurring.

## 4.1 RESULTS

**Zero-shot Image Classification.** We follow the standard evaluation benchmark and made sure all prompts and class names were the same as those used by CLIP Radford et al. (2021). We also re-evaluated OpenAI/OpenCLIP's checkpoints to avoid differences caused by benchmark data copies. The results are shown in Tab 4. The standard deviation of training multiple seeds is relatively small with ± 0.1% for ImageNet on ViT-B/32.

In Table 4, we observe that MetaCLIP outperforms OpenAI CLIP on ImageNet and average accuracy across 26 tasks, for 3 model scales. With 400 million training data points on ViT-B/32, MetaCLIP outperforms CLIP by +2.1% on ImageNet and by +1.6% on average. On ViT-B/16, MetaCLIP outperforms CLIP by +2.5% on ImageNet and by +1.5% on average. On ViT-L/14, MetaCLIP outperforms CLIP by +0.7% on ImageNet and by +1.4% on average across the 26 tasks.

We next turn to Pool 2 which is a larger set of image-text pairs and study the effect of scaling data. In Table 5, we scale data to 1B and 2.5B and observe a large gain over 400M, with similar performance for both 1B and 2.5B scales. Note that the number of training iterations (and therefore compute) is the same for all rows. The main difference between 1B and 2.5B is the threshold $t$, where 1B is a more balanced set by adding more data points (compared to the 400M set) to *tail* entries (up to $t = 20k$), instead the 2.5B set adds (up to $t = 170k$) data points to all, *head and tail*, entries. The extra data in the tail entries (1B set), seems to benefit downstream accuracy for tasks on specific data such as CUB fine-grained bird classification, Flowers, KITTI, PCAM, while the larger 2.5B data that has more head entries increases broadly over more datasets, but each at a smaller amount. The overall average accuracies are similar for 1B and 2.5B (e.g., 70.2% vs. 69.8% for ViT-L model

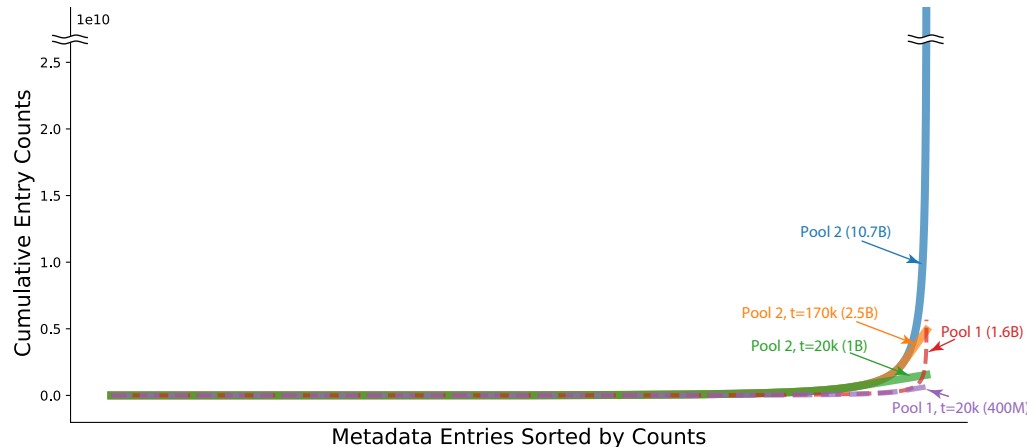

Figure 3: Cumulative sum of counts on entries from *tail to head* on a data Pool 2. We again show (1) raw/unbalanced cumulative counts), $t = \infty$; (2) balanced cumulative counts after applying $t = 20k$ and $t = 170k$. $t$ defines maximum number of counts per entry and the transition of tail/head entries. We show the Pool 1 configuration from Fig. 2 as dashed lines for reference.

size). On ImageNet, the 2.5B training data achieves 67.6% on ViT-B/32 that breaks the previous believed saturated B/32 models (Cherti et al., 2022), 79.2% on ViT-L/14 , 80.5% on ViT-H/14 and 82.1% on ViT-bigG/14.

We plot the cumulative sum of counts for entries sorted by counts from tail to head in Fig. 3 for all these cases, similar to Fig. 2 for Pool 1 (and the Pool 1 configuration as dashed lines). The plot shows that the 2.5B data is still relatively long-tail, while the 1B data is more balanced, explaining it's better performance on specific data such as bird and flower types observed above.

## 4.2 ABLATION STUDY

We show ablations for MetaCLIP for the 400M scale and ViT-B/32 in Table 6. We first ablate different balancing thresholds $t$. We observe that the choice of $t = 20k$ by CLIP yields the best performance for ImageNet and averaged accuracy and $t = 15k$ or $t = 35k$ are slightly worse.

To understand the key effect of balancing, we use the whole matched pool (1.6B image-text pairs) to train CLIP. Surprisingly, training on $4\times$ *more data* (on head entries) *significantly hurts the accuracy* on ImageNet (61.9 vs 65.5) and averaged accuracy across 26 tasks (56.6 vs 58.2).

Balancing can also be applied online in the data loader with head entries down-sampled leading to slightly better performance (58.5 vs 58.2); see appendix for details. This is useful if head data has already been collected and one wants to train on a different distribution. The better accuracy for online balancing is explained by the larger diversity in head data.

| | Average | ImageNet | Food-101 | CIFAR10 | CIFAR100 | CUB | SUN397 | Cars | Aircraft | DTD | Pets | Caltech-101 | Flowers | MNIST | FER-2013 | STL-10 | EuroSAT | RESISC45 | GTSRB | KITTI | Country211 | PCAM | UCF101 | Kinetics700 | CLEVR | HatefulMemes | SST2 |
|---|---|---|---|---|---|---|---|---|---|---|---|---|---|---|---|---|---|---|---|---|---|---|---|---|---|---|---|
| MetaCLIP $t$=20k | 58.2 | 65.5 | 80.6 | 91.3 | 70.2 | 63.4 | 63.0 | 70.7 | 26.8 | 52.8 | 88.7 | 91.9 | 68.5 | 41.5 | 35.9 | 95.4 | 52.6 | 64.2 | 35.8 | 30.7 | 17.2 | 55.5 | 66.1 | 45.4 | 30.6 | 56.4 | 53.4 |
| - $t$=15k | 57.5 | 65.5 | 79.9 | 90.4 | 68.8 | 65.7 | 64.6 | 69.4 | 25.6 | 52.1 | 88.8 | 91.9 | 69.5 | 35.8 | 39.7 | 96.5 | 54.0 | 64.1 | 34.8 | 30.6 | 16.1 | 52.3 | 67.1 | 45.4 | 22.3 | 51.2 | 53.8 |
| - $t$=35k | 57.8 | 65.4 | 79.3 | 91.2 | 69.0 | 63.0 | 65.0 | 72.0 | 28.5 | 52.7 | 88.5 | 91.8 | 68.0 | 42.0 | 23.0 | 96.2 | 50.0 | 63.8 | 40.2 | 32.4 | 17.7 | 56.1 | 64.2 | 44.8 | 28.0 | 55.4 | 54.2 |
| - unbalanced (1.6B) | 56.6 | 61.9 | 76.9 | 90.0 | 67.6 | 50.8 | 65.8 | 77.0 | 19.9 | 51.0 | 83.1 | 91.5 | 64.5 | 58.2 | 37.0 | 95.1 | 55.2 | 58.2 | 41.4 | 32.2 | 15.1 | 51.0 | 59.2 | 42.6 | 17.2 | 55.6 | 52.6 |
| - online balancing | 58.5 | 66.1 | 80.8 | 89.9 | 68.8 | 65.7 | 65.4 | 71.6 | 27.9 | 55.1 | 88.2 | 92.7 | 68.8 | 38.3 | 42.1 | 96.5 | 54.5 | 64.8 | 36.2 | 29.1 | 17.6 | 58.8 | 66.0 | 45.8 | 22.0 | 56.0 | 52.4 |

Table 6: Ablation studies on balancing in MetaCLIP. Default: $t$=20k, 400M. Model: ViT-B/32.

## 5 CONCLUSION

In this paper, we attempt to reveal CLIP's data curation. Our MetaCLIP builds upon metadata for curation and balancing of raw data sourced from the web. Curating with metadata and balancing are essential for good data quality, significantly outperforming the use of raw data. Our experiments show that MetaCLIP performs well for different scales sourced from CommonCrawl data and outperforms CLIP's proprietary data source, without reliance on any external model. We make our pipeline for generating the data publicly available.

ACKNOWLEDGMENTS

We thank Zeyuan Allen-Zhu, and Chunting Zhou for the insightful discussion and Brighid Meredith for suggestions on scaling the pipeline.

## A  APPENDIX

### A.1  ADDITIONAL RESULTS

**Curation from DataComp-12.8B.** The concurrent work Gadre et al. (2023) released a collection of 12.8B image-text pairs from CommonCrawl from 2014-2022. We further investigate whether we can apply the algorithm on its 12.8 unfiltered pool. Although the unfiltered pool seemingly offers an opportunity to apply our algorithm on a publicly available source, our initial studies show that, implicit biases may still be present in this pool. For example, we notice that all image URLs are collected as a string starting with `http`. This excludes relative URLs that could be frequently used by quality websites (with potentially good image-text pairs). We curate from Data-Comp's 12.8B unfiltered pool with $t=60k$, which has 6% of tail counts that is the same as $t=20k$ for 400M from our 1.6B pool.

When using 1B image-text pairs curated from DataComp's pool we notice a quality drop during training, compared to data curated from our pools, see Fig. 4. Our smaller 400M set is slightly better than using DataComp-1B and our larger sets (1B, 2.5B) are significantly better.

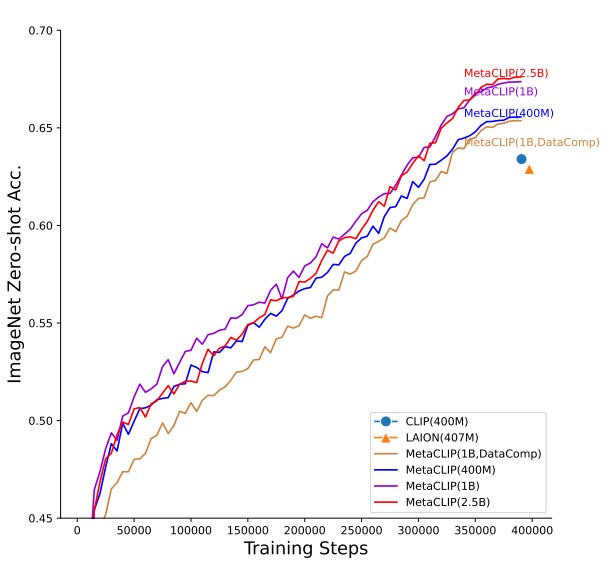

Figure 4: ViT-B/32 on our Pool 1, Pool 2 and DataComp's unfiltered 12.8B pool. We show ImageNet zero-shot classification with a fixed 12.8B seen pair training budget. Meta-CLIP's curation is effective for all pools. However, with the same curation method, the unfiltered DataComp-12.8B pool lacks quality (we suspect it is caused by implicit filters in the parser of DataComp).

In Table 7, we show our 400M data vs our curated DataComp-1B data at various model scales, where the same observation holds, suggesting our raw data pool is more effective.

| | Average | ImageNet | Food-101 | CIFAR10 | CIFAR100 | CUB | SUN397 | Cars | Aircraft | DTD | Pets | Caltech-101 | Flowers | MNIST | FER-2013 | STL-10 | EuroSAT | RESISC45 | GTSRB | KITTI | Country211 | PCAM | UCF101 | Kinetics700 | CLEVR | HatefulMemes | SST2 |
|---|---|---|---|---|---|---|---|---|---|---|---|---|---|---|---|---|---|---|---|---|---|---|---|---|---|---|---|
| **ViT-B/32** | | | | | | | | | | | | | | | | | | | | | | | | | | | |
| MetaCLIP (400M) | 58.2 | 65.5 | 80.6 | 91.3 | 70.2 | 63.4 | 63.0 | 70.7 | 26.8 | 52.8 | 88.7 | 91.9 | 68.5 | 41.5 | 35.9 | 95.4 | 52.6 | 64.2 | 35.8 | 30.7 | 17.2 | 55.5 | 66.1 | 45.4 | 30.6 | 56.4 | 53.4 |
| MetaCLIP (1B,DataComp) | 57.5 | 65.4 | 81.5 | 88.7 | 68.5 | 59.4 | 64.7 | 76.6 | 18.1 | 57.5 | 89.9 | 92.4 | 67.3 | 40.7 | 38.2 | 96.5 | 41.2 | 62.3 | 43.4 | 35.1 | 17.6 | 53.5 | 61.5 | 44.8 | 19.2 | 57.0 | 54.3 |
| **ViT-B/16** | | | | | | | | | | | | | | | | | | | | | | | | | | | |
| MetaCLIP (400M) | 61.1 | 70.8 | 86.8 | 90.1 | 66.5 | 70.8 | 66.6 | 74.1 | 27.9 | 55.9 | 90.4 | 93.8 | 72.3 | 47.8 | 44.6 | 97.2 | 55.4 | 68.8 | 43.8 | 33.4 | 22.6 | 52.9 | 68.0 | 49.5 | 22.8 | 54.8 | 60.6 |
| MetaCLIP (1B,DataComp) | 61.2 | 70.7 | 88.1 | 91.3 | 71.6 | 64.5 | 67.6 | 81.2 | 21.6 | 62.3 | 92.7 | 93.2 | 70.7 | 55.6 | 39.6 | 97.7 | 52.9 | 66.3 | 45.7 | 36.0 | 22.3 | 50.1 | 68.1 | 49.2 | 17.1 | 56.4 | 59.9 |
| **ViT-L/14** | | | | | | | | | | | | | | | | | | | | | | | | | | | |
| MetaCLIP (400M) | 67.1 | 76.2 | 90.7 | 95.5 | 77.4 | 75.9 | 70.5 | 84.7 | 40.4 | 62.0 | 93.7 | 94.4 | 76.4 | 61.7 | 46.5 | 99.3 | 59.7 | 71.9 | 47.5 | 29.9 | 30.9 | 70.1 | 75.5 | 57.1 | 35.1 | 56.6 | 65.6 |
| MetaCLIP (1B,DataComp) | 66.3 | 76.7 | 92.6 | 95.6 | 78.9 | 72.1 | 71.6 | 87.1 | 31.6 | 67.5 | 93.4 | 95.3 | 76.0 | 65.1 | 42.4 | 99.1 | 61.7 | 69.8 | 45.9 | 36.8 | 31.8 | 51.0 | 76.6 | 57.5 | 29.3 | 57.1 | 60.8 |

Table 7: MetaCLIP curating our 400M data vs curating 1B data from DataComp-12.8B: The pool of DataComp leads to a quality drop with performance closer to our 400M set.

**DataComp Benchmark**  We also evaluate MetaCLIP on the benchmark used by (Gadre et al., 2023) that contains 38 tasks including variants of ImageNet, retrieval, VTAB, etc. For simplicity, we average the scores over each category.

|  | Avg. | IN | IN Dist. Shift | VTAB | Avg. Retrieval |
|---|---|---|---|---|---|
| **ViT-B/32** | | | | | |
| CLIP (400M) | 51.5 | 63.4 | 48.2 | 50.5 | 48.0 |
| OpenCLIP (407M) | 52.7 | 62.9 | 48.5 | 53.0 | 50.7 |
| MetaCLIP (400M) | 53.5 | 65.5 | 50.4 | 54.1 | 50.6 |
| MetaCLIP (1B) | 54.2 | 67.3 | 51.9 | 53.6 | 51.1 |
| MetaCLIP (2.5B) | 55.4 | 67.6 | 52.3 | 55.3 | 52.6 |
| | | | | | |
| **ViT-B/16** | | | | | |
| CLIP (400M) | 55.5 | 68.3 | 54.1 | 54.4 | 50.2 |
| OpenCLIP (407M) | 56.1 | 67.0 | 52.6 | 54.9 | 53.9 |
| MetaCLIP (400M) | 57.5 | 70.8 | 55.5 | 56.7 | 53.9 |
| MetaCLIP (1B) | 58.4 | 72.4 | 57.8 | 56.3 | 54.3 |
| MetaCLIP (2.5B) | 60.0 | 72.1 | 57.7 | 59.0 | 54.0 |
| | | | | | |
| **ViT-L/14** | | | | | |
| CLIP (400M) | 61.4 | 75.5 | 61.6 | 59.5 | 53.6 |
| OpenCLIP (407M) | 59.7 | 72.7 | 57.3 | 58.6 | 55.9 |
| MetaCLIP (400M) | 62.2 | 76.2 | 61.3 | 59.8 | 57.3 |
| MetaCLIP (1B) | 65.0 | 79.0 | 64.5 | 62.5 | 58.3 |
| MetaCLIP (2.5B) | 65.6 | 79.2 | 64.6 | 64.1 | 60.1 |
| | | | | | |
| **ViT-H/14** | | | | | |
| MetaCLIP (2.5B) | 66.5 | 80.5 | 66.1 | 64.6 | 60.4 |
| | | | | | |
| **ViT-bigG/14** | | | | | |
| MetaCLIP (2.5B) | 68.3 | 82.1 | 67.6 | 66.5 | 62.6 |

Table 8: Zero-shot classification and retrieval on tasks from (Gadre et al., 2023).

Note that prompts and class names used by (Gadre et al., 2023) could be different from prompts and classnames used by OpenAI CLIP.

From Table 8, we can see that MetaCLIP outperforms CLIP and OpenCLIP across various model sizes. First, for the same data scale (400M), MetaCLIP outperforms OpenCLIP, which is better than CLIP on this benchmark, by +1.4% for ViT-B/16 and +2.5% for ViT-L/14, when comparing average accuracy across the 38 tasks. Second, for increasing our MetaCLIP data size to 1B we see a significant gain, especially for the larger model, from 62.2% to 65.0% average accuracy. Using our larger dataset with 2.5B and more head entries leads to a further gain to 65.5%.

We further detail the breakdown of each dataset in Table 9 (please view in landscape orentation).

Table 9: Zero-shot evaluation of Table 8 broken down by datasets.

| Model | ImageNet 1k | ImageNet Sketch | ImageNet V2 | ImageNet-A | ImageNet-O | ImageNet-R | Caltech-101 | CIFAR-100 | CLEVR Counts | CLEVR Distance | Describable Textures | EuroSAT | KITTI Vehicle Distance | Oxford Flowers-102 | Oxford-IIIT Pet | RESISC45 | SUN397 | SVHN | Camelyon17 | Flickr | MSCOCO | WinoGAViL | CIFAR-10 | Country211 | FGVC Aircraft | Food-101 | GTSRB | MNIST | ObjectNet | Pascal VOC 2007 | PatchCamelyon | Rendered SST2 | Stanford Cars | STL-10 | iWildCam | FMoW | Dollar Street | GeoDE |
|---|---|---|---|---|---|---|---|---|---|---|---|---|---|---|---|---|---|---|---|---|---|---|---|---|---|---|---|---|---|---|---|---|---|---|---|---|---|---|
| **ViT-B/32** | | | | | | | | | | | | | | | | | | | | | | | | | | | | | | | | | | | | | | |
| CLIP (400M) | 63.3 | 42.3 | 56.0 | 31.5 | 47.8 | 69.4 | 87.6 | 64.2 | 23.3 | 23.4 | 44.3 | 50.5 | 27.4 | 66.6 | 87.0 | 53.6 | 62.5 | 14.9 | 51.6 | 68.8 | 40.3 | 34.9 | 89.8 | 17.2 | 19.5 | 84.0 | 32.5 | 48.3 | 44.3 | 76.4 | 62.2 | 58.6 | 59.6 | 97.1 | 6.7 | 13.3 | 53.9 | 82.2 |
| OpenCLIP (407M) | 62.9 | 49.4 | 55.1 | 21.7 | 53.5 | 73.4 | 91.2 | 70.3 | 16.2 | 23.9 | 54.6 | 51.5 | 28.8 | 66.2 | 86.7 | 54.5 | 67.0 | 30.4 | 47.1 | 70.2 | 43.9 | 38.0 | 90.7 | 14.7 | 16.6 | 80.9 | 42.0 | 37.5 | 43.9 | 75.8 | 55.9 | 52.3 | 79.3 | 95.6 | 7.4 | 13.0 | 54.9 | 83.8 |
| MetaCLIP (400M) | 65.6 | 53.3 | 57.6 | 28.6 | 46.8 | 74.8 | 91.7 | 70.0 | 21.5 | 24.5 | 52.5 | 52.4 | 25.9 | 68.1 | 88.8 | 59.6 | 63.4 | 20.5 | 64.5 | 70.1 | 43.9 | 37.8 | 91.3 | 17.1 | 26.9 | 81.0 | 35.8 | 41.4 | 50.5 | 70.8 | 64.2 | 53.5 | 70.6 | 95.4 | 8.2 | 0.0 | 59.7 | 85.4 |
| MetaCLIP (1B) | 67.3 | 55.3 | 59.2 | 29.7 | 48.2 | 76.1 | 93.7 | 76.5 | 15.7 | 22.5 | 59.1 | 48.0 | 20.5 | 72.7 | 89.5 | 61.6 | 66.3 | 21.7 | 49.4 | 72.6 | 45.1 | 35.6 | 95.2 | 16.9 | 31.0 | 82.6 | 40.7 | 35.6 | 49.4 | 79.0 | 50.0 | 57.1 | 73.0 | 96.3 | 8.4 | 14.8 | 58.9 | 86.0 |
| MetaCLIP (2.5B) | 67.7 | 56.0 | 59.6 | 29.9 | 48.3 | 78.1 | 92.9 | 77.7 | 18.7 | 23.1 | 58.8 | 49.9 | 18.7 | 69.4 | 90.9 | 61.2 | 66.9 | 34.3 | 56.6 | 72.9 | 46.6 | 38.2 | 95.2 | 17.7 | 27.1 | 83.1 | 39.3 | 42.6 | 52.8 | 76.5 | 56.0 | 53.2 | 77.3 | 96.3 | 9.2 | 15.9 | 60.5 | 86.1 |
| **ViT-B/16** | | | | | | | | | | | | | | | | | | | | | | | | | | | | | | | | | | | | | | |
| CLIP (400M) | 68.3 | 48.3 | 61.9 | 49.9 | 42.3 | 77.7 | 89.0 | 66.9 | 21.2 | 22.3 | 44.9 | 56.0 | 26.3 | 69.1 | 88.9 | 58.2 | 64.4 | 40.2 | 60.2 | 72.2 | 42.8 | 35.7 | 90.8 | 22.8 | 24.3 | 88.7 | 43.3 | 51.3 | 55.3 | 78.3 | 50.7 | 60.5 | 64.7 | 98.3 | 11.1 | 16.7 | 58.8 | 86.1 |
| OpenCLIP (407M) | 67.0 | 52.4 | 59.7 | 33.2 | 50.6 | 77.9 | 91.3 | 71.2 | 28.7 | 24.5 | 51.3 | 50.2 | 18.1 | 66.9 | 89.2 | 58.5 | 69.6 | 34.1 | 59.9 | 74.5 | 46.9 | 40.2 | 91.7 | 18.1 | 17.7 | 86.1 | 43.4 | 66.2 | 51.5 | 76.8 | 59.6 | 54.4 | 83.7 | 97.0 | 10.3 | 15.5 | 59.3 | 85.3 |
| MetaCLIP (400M) | 70.8 | 57.9 | 62.6 | 47.0 | 39.2 | 81.8 | 93.4 | 66.5 | 30.1 | 22.4 | 55.7 | 55.7 | 24.2 | 72.3 | 90.4 | 66.1 | 66.8 | 25.2 | 67.7 | 76.7 | 48.2 | 36.9 | 90.1 | 22.6 | 28.4 | 87.2 | 43.7 | 47.8 | 59.1 | 72.2 | 61.9 | 60.5 | 74.2 | 97.2 | 11.2 | 19.9 | 60.6 | 88.9 |
| MetaCLIP (1B) | 72.4 | 60.5 | 65.1 | 49.2 | 41.9 | 82.7 | 93.5 | 77.9 | 19.3 | 24.4 | 58.2 | 47.6 | 24.6 | 74.8 | 92.1 | 65.1 | 66.8 | 27.8 | 60.1 | 77.1 | 48.9 | 36.8 | 94.8 | 21.6 | 38.1 | 88.3 | 49.5 | 36.3 | 60.0 | 79.2 | 65.7 | 61.3 | 79.0 | 98.0 | 13.2 | 15.8 | 63.3 | 87.9 |
| MetaCLIP (2.5B) | 72.1 | 60.2 | 65.0 | 49.5 | 41.5 | 84.2 | 93.3 | 78.9 | 29.0 | 22.6 | 62.2 | 52.7 | 18.4 | 73.5 | 91.7 | 67.4 | 68.8 | 39.1 | 69.8 | 78.1 | 50.4 | 33.5 | 95.7 | 22.7 | 30.5 | 88.8 | 46.5 | 66.1 | 61.4 | 78.2 | 59.2 | 59.1 | 83.0 | 98.4 | 12.3 | 19.4 | 64.0 | 88.7 |
| **ViT-L/14** | | | | | | | | | | | | | | | | | | | | | | | | | | | | | | | | | | | | | | |
| CLIP (400M) | 75.6 | 59.6 | 69.8 | 70.7 | 32.3 | 87.9 | 92.5 | 75.8 | 19.5 | 20.2 | 55.3 | 62.6 | 21.8 | 79.2 | 93.2 | 63.3 | 67.6 | 52.8 | 69.6 | 75.1 | 46.4 | 39.2 | 95.6 | 31.9 | 31.7 | 93.1 | 50.6 | 76.3 | 68.9 | 78.3 | 52.0 | 68.9 | 77.9 | 99.4 | 12.3 | 15.2 | 63.0 | 88.4 |
| OpenCLIP (407M) | 72.8 | 59.6 | 65.4 | 46.5 | 41.9 | 84.7 | 92.6 | 77.4 | 24.3 | 24.5 | 60.5 | 62.3 | 20.1 | 73.1 | 91.7 | 67.4 | 72.6 | 49.5 | 45.5 | 78.9 | 51.3 | 37.4 | 94.6 | 23.0 | 24.9 | 90.1 | 49.9 | 76.1 | 59.7 | 75.6 | 49.7 | 56.0 | 89.6 | 98.1 | 12.5 | 17.0 | 61.7 | 88.4 |
| MetaCLIP (400M) | 76.2 | 65.0 | 69.8 | 66.4 | 28.9 | 88.9 | 94.6 | 77.3 | 22.7 | 25.1 | 62.4 | 60.4 | 24.2 | 76.5 | 93.8 | 68.5 | 70.8 | 32.4 | 69.2 | 79.8 | 51.9 | 40.2 | 95.5 | 30.9 | 39.8 | 90.7 | 47.5 | 61.8 | 69.2 | 74.4 | 70.3 | 65.6 | 84.8 | 99.3 | 14.1 | 18.7 | 67.3 | 89.3 |
| MetaCLIP (1B) | 79.0 | 68.9 | 72.5 | 70.4 | 31.6 | 91.0 | 98.4 | 84.6 | 15.8 | 22.5 | 65.8 | 68.3 | 23.2 | 83.8 | 95.2 | 68.0 | 72.9 | 38.0 | 79.1 | 82.2 | 54.2 | 38.6 | 96.8 | 32.2 | 49.6 | 93.1 | 58.6 | 53.9 | 73.9 | 80.9 | 73.1 | 66.0 | 86.6 | 99.3 | 16.2 | 28.1 | 69.4 | 91.1 |
| MetaCLIP (2.5B) | 79.2 | 68.9 | 72.6 | 72.3 | 30.1 | 92.0 | 95.3 | 84.1 | 31.0 | 22.6 | 68.6 | 59.0 | 27.9 | 81.4 | 94.6 | 73.6 | 73.6 | 46.8 | 75.5 | 83.3 | 55.7 | 41.4 | 97.6 | 33.9 | 45.3 | 93.5 | 56.3 | 64.4 | 74.6 | 80.3 | 62.0 | 66.8 | 88.8 | 99.3 | 15.8 | 25.9 | 67.4 | 91.4 |
| **ViT-H/14** | | | | | | | | | | | | | | | | | | | | | | | | | | | | | | | | | | | | | | |
| MetaCLIP (2.5B) | 80.5 | 70.5 | 74.1 | 75.4 | 30.2 | 93.4 | 95.3 | 86.4 | 21.1 | 18.8 | 72.7 | 64.5 | 27.7 | 84.5 | 95.6 | 70.3 | 74.4 | 62.8 | 66.2 | 85.0 | 57.5 | 38.8 | 98.1 | 37.2 | 51.0 | 94.2 | 62.6 | 72.7 | 76.5 | 75.0 | 62.3 | 69.1 | 89.9 | 99.4 | 17.3 | 15.6 | 68.1 | 90.7 |
| **ViT-bigG/14** | | | | | | | | | | | | | | | | | | | | | | | | | | | | | | | | | | | | | | |
| MetaCLIP (2.5B) | 82.1 | 72.8 | 76.0 | 78.4 | 28.9 | 94.1 | 95.7 | 88.3 | 17.0 | 20.3 | 72.1 | 72.8 | 26.7 | 89.1 | 96.2 | 74.0 | 75.1 | 60.4 | 76.1 | 85.4 | 58.1 | 44.4 | 98.5 | 41.5 | 52.7 | 94.8 | 61.7 | 78.1 | 77.2 | 74.7 | 75.8 | 65.3 | 90.8 | 99.5 | 18.3 | 18.5 | 70.2 | 92.5 |

A.2    DETAILS ON EFFICIENT CURATION

**Curation in Data Pipeline.**    Our data collection pipeline contains 5 major components: HTML parsing/Language-Identification, URLs & Text deduplication, Image Download, NSFW image filter & deduplication and packaging, which are described next.

Our HTML Parser is applied to all WAT files of CommonCrawl by keeping all `` tags with both relative and absolute URLs and one or multiple text keys.

Language-Identification(LID): we use an internal language identification system that can detect 191 languages with different dialects and keep all texts that are tagged as English (or its dialects).

URLs / Text deduplication: We use 24bit sha224 hashing to encode an URL and into tables to deduplicate columns with the same hash, avoiding downloading the same image URL multiple times. URLs with illegal domains are removed; if there are multiple texts associated with the same image, these are further deduplicated; texts with NSFW keywords are removed;

NSFW filter: We use an internal system that can classify inappropriate content in images into 96 types of dangerous content and discard such image/text pairs. Images are further deduplicated by 64-bit PCA hash, derived from a similarity search model's feature embeddings with PCA reduction to 64 dimensions and sign quantization.

Our curation algorithm does not require access to images, making it suitable for integration into a pipeline to reduce the scale of data points after parsing and before image downloading. We designed the algorithm to be modular, allowing different parts to be placed at different stages in the pipeline, as shown in Figure 5.

Specifically, sub-string matching can be placed immediately after HTML parsing to reduce data points for English-only pairs (e.g. by ∼50%). Balancing can be applied earlier, before image downloading, to further reduce data points by ∼77%. This approach led to a total reduction of ∼90%, allowing for curation of the entire CommonCrawl data *without* the need to store and transfer all data points, which allows us to curate the whole CommonCrawl since 2013 with 300B+ scale URL-text pairs without spending the storage/transfer on all ∼10× data points, where the rate of keeping a data point with MetaCLIP curation is ∼0.1 ($0.5 \times 0.23$).

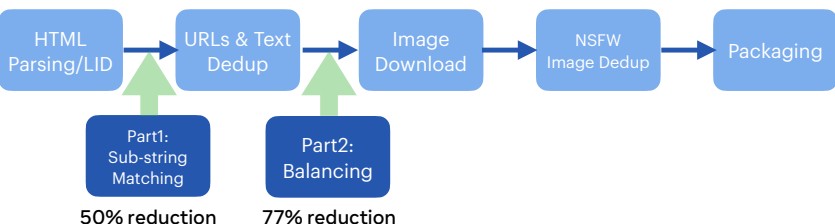

Figure 5: Case study: Curation implementation in our data pipeline.

**Curation in Data Loading**    We applied the balancing/sampling part of the algorithm to the data loader to adjust data distribution on-the-fly. Although data points for tail entries are always sampled in Algorithm 1, the diverse pairs from the head entries are sub-sampled from a larger pool while maintaining a similar distribution as offline curation. This diversification of pairs that match the head entries improved the performance, as shown in Table 6.

A.3    HUMAN STUDY ON THE EFFECTS OF CURATION

In this section, we study the impact of MetaCLIP curation on data distribution by using human evaluation. We approach this exploration from three distinct angles: noise reduction, alignment of visual content, and task-agnostic attributes. In the pursuit of comprehending the first two aspects, we undertake a human study aimed at comprehending the data quality implications of implementing the balancing technique (outlined in Part 2 of Algorithm 1). This evaluation encompasses three dimensions: image-only, text-only, and image-text alignment. We collect an evaluation set of 100 random image-text pairs for balanced and unbalanced data, respectively, and ask annotators to score on the image, text, and pair quality metrics, separately, on a scale of 1 to 5.

**Annotation Guidelines.** Annotators follow guidelines to assess both images and texts, evaluating informativeness (how well information is conveyed) and aesthetics. For images, aesthetics considers visual elements like composition, color, lighting, balance, contrast, texture, and subject matter. For texts, aesthetics gauges factors like delimiters, sentence structure, capitalization, prefixes/suffixes, recognized words, generic words, and overall text quality. The alignment metric for image-text pairs measures the relevance between the two modalities, assessing how well the text describes the image content. Ratings are averaged across annotators for each dimension.

We show the study results in Table 10 and discuss the different criteria next.

| Evaluation Dimension | Rating for Balanced Data | Rating for Unbalanced Data | P-value |
|:---:|:---:|:---:|:---:|
| Image | 4.60, [4.50, 4.70] | 4.36, [4.23, 4.48] | < 0.001 |
| Text | 4.67, [4.56, 4.78] | 4.06, [3.82, 4.30] | < 0.001 |
| Alignment | 4.41, [4.23, 4.59] | 3.72, [3.46, 3.99] | < 0.001 |

Table 10: Average human rating on the effect of balancing on data quality, with confidence intervals shown in parentheses. Higher rating is better. Balanced data is rated of higher quality.

**Noise Mitigation in Image and Text.** As shown in Table 10, significant quality improvement for all the three evaluation dimensions is observed after applying balancing. MetaCLIP curation has no specific hard filters such as removing shorter text, removing dates, etc. However, curation by sub-string matching and balancing has a different filtering effect. For example, a sub-string itself can never curate a date-only text. Further, balancing allows signal and noise to co-exist when they are difficult to be separated by human designed filters. For example, if one entry such as "image" or "photo" is capped to $t = 20k$, it can only contribute 0.005% of 400M data.

**Visual Content Alignment.** Although MetaCLIP curation does not directly involve images, it has a positive effect on aligning visual content by controlling the quality and distribution of text. First, sub-string matching increases the chance of having (visual) entities mentioned in the text, thereby improving the likelihood of finding corresponding visual content. Second, balancing favors long-tailed entries that could have more diverse visual content than a head entry (such as the text "1"). In Table 10, we observe significant improvement on pair quality from unbalanced data to balanced data.

## A.4 MEASURING TASK-ALIGNMENT

With metadata, one can measure the alignment of pre-training data distribution with the data distribution in downstream tasks. First we can do a simple measure of counting the number of metadata substring matches that are directly corresponding to class names in the downstream tasks. In the first row of Table 11, we show the accuracy of MetaCLIP (400M), ViT-L (*cf.* Table4) for reference. The second row shows the number of classes that are present in the metadata, for each downstream dataset. For example, for ImageNet 703 of the 998 unique class names are present in the metadata. Interestingly, there seems to be a correlation with the accuracy and the number of classes matched in the metadata.

| | ImageNet | Food-101 | CIFAR10 | CIFAR100 | CUB | SUN397 | Cars | Aircraft | DTD | Pets | Caltech-101 | Flowers | MNIST | FER-2013 | STL-10 | EuroSAT | RESISC45 | GTSRB | KITTI | Country211 | PCAM | UCF101 | Kinetics700 | CLEVR | HatefulMemes | SST2 |
|---|---|---|---|---|---|---|---|---|---|---|---|---|---|---|---|---|---|---|---|---|---|---|---|---|---|---|
| MetaCLIP (400M) ViT-L | 76.2 | 90.7 | 95.5 | 77.4 | 75.9 | 70.5 | 84.7 | 40.4 | 62.0 | 93.7 | 94.4 | 76.4 | 61.7 | 46.5 | 99.3 | 59.7 | 71.9 | 47.5 | 29.9 | 30.9 | 70.1 | 75.5 | 57.1 | 35.1 | 56.6 | 65.6 |
| # of cls. w/ non-zero counts | 703/998 | 52/101 | 10/10 | 93/100 | 1/200 | 193/397 | 0/196 | 8/100 | 40/47 | 15/37 | 86/102 | 61/102 | 10/10 | 12/12 | 10/10 | 2/10 | 32/45 | 1/43 | 0/4 | 190/211 | 1/2 | 5/101 | 122/700 | 8/8 | 1/2 | 2/2 |
| KL-divergence | | | | | | | | | | | | | | | | | | | | | | | | | | |
| - unbal. | 6.8 | 9.4 | 7.5 | 6.1 | 11.4 | 6.7 | 0.0 | 12.0 | 9.7 | 11.7 | 7.7 | 10.6 | 3.4 | 8.9 | 7.1 | 8.4 | 7.6 | 9.3 | 0.0 | 5.7 | 14.6 | 8.9 | 8.9 | 3.8 | 9.2 | 9.8 |
| - bal. | 5.1 | 7.4 | 8.1 | 6.0 | 10.4 | 6.0 | 0.0 | 10.1 | 8.0 | 9.7 | 7.1 | 8.6 | 8.1 | 8.3 | 8.1 | 9.7 | 7.2 | 10.4 | 0.0 | 5.2 | 12.5 | 8.8 | 7.1 | 8.3 | 10.4 | 9.7 |

Table 11: Measuring task-alignment. First row: MetaCLIP (400M) ViT-L/14 accuracy, second row: number of classes matched in metadata, Third and fourth row: We use KL-divergence (lower is better) to measure the similarity between pre-training distribution and benchmark task distribution. Note that, for ImageNet, two classes have duplicated class names and therefore there are only 998 unique classes (out of 1000).

| Hyperparameter | OpenAI CLIP / MetaCLIP | OpenCLIP | DataComp |
|---|---|---|---|
| Activation Function | QuickGELU | GELU | GELU |
| Seen Pairs | 12.8B(400M×32 epochs) | 13B (407M×32 epochs) | 12.8B |
| Batch Size | 32768 | 32768 (B/32), 33792 (B/16), 38400 (L/14) | 90112 (L/14) |
| Learning Rate | 5.0e-4(B/32,B/16), 4.0e-4(L/14) | 5.0e-4(B/32) | 1e-3(L/14) |
| Warm-up | 2k | 2k (B/32) | 10k (L/14) |

Table 12: Hyperparameters of OpenAI CLIP vs OpenCLIP on LAION-400M(Schuhmann et al., 2021) and DataComp 1B.

Next, we use KL-divergence to measure alignment:

$$D_{\text{KL}}(T||P) = - \sum_{m \in \mathcal{M}} T(m) \log \big( \frac{P(m)}{T(m)} \big), \tag{1}$$

where $T(m)$ represents the task distribution over $\mathcal{M}$ and $P(m)$ represents the pre-training data distribution. We compute $T(m)$ using the benchmark data distribution over class labels of a task. Taking ImageNet as an example from Table 11, $T(m)$ ImageNet has 998 entries uniformly distributed (each has evenly 50 examples) and $P(m)$ has normalized counts over all 500k entries but only 703 entries used to compute KL-divergence. In the third and fourth row of Table 11, we can see improvements for balanced data points over unbalanced ones for each task, showing balancing improves similarity with most tasks.

## A.5 TRAINING SETUP OF OPENAI CLIP VS OPENCLIP

Our work strictly follows CLIP's setup for a controlled comparison focusing on data curation and quality. We notice differences in the training setup of OpenCLIP[5] and list the difference (known to us). OpenCLIP varies the setup from CLIP (e.g., global batch size, learning schedule, etc.). Here we only list the difference for LAION-400M(Schuhmann et al., 2021), which is closer to the CLIP setup. We note that DataComp differs even more, e.g., by curating images close to ImageNet training data, a large batch size of 90k that is almost $3\times$ larger than CLIP, and using the CLIP model to filter data.

## A.6 BENCHMARK DEDUPLICATION

Our pools are deduplicated from the benchmark/ImageNet data using a 64-bit PCA hash, derived from a similarity search model's feature embeddings with PCA reduction to 64 dimensions and sign quantization. DataComp-12.8B is already deduplicated.

## A.7 NEGATIVE RESULTS LEARNED FROM ABLATING CLIP CURATION

We briefly describe a few ideas close to CLIP curation that did not look promising in our initial attempts and were abandoned:

1. **Self-curated Metadata**. We initially attempted to build metadata directly from the text in raw image-text pairs (i.e., using terms appearing in text above a certain threshold of counts). We rank entries by count and keep the top 500,000. Metadata built this way appeared worse. We notice that although the top frequent entries are similar to CLIP's metadata, the long-tailed part is very different. For example, the minimal count to be in the 500,000 budget needs at least 130 counts. In contrast, our metadata has 114K entries that have no matches. This approach results in worse quality metadata including low-quality spelling/writing (instead of high-quality entries from WordNet or Wikipedia). Further, the effect of balancing saturates earlier for such data (in a larger $t$, verified by CLIP training) since low-quality entries are also heavily in long-tail.

2. **Cased WordNet**. We also notice many cased words are missing from metadata (e.g., WordNet is in lowercase). After adding cased WordNet into metadata, we notice a performance drop on ImageNet. The reason could be class names are more likely in lower case and upper case entry matching may reduce the written quality of texts.

---

[5]https://github.com/mlfoundations/open_clip

3. **Stopwords/Useless Entries Removal** We further study the effect of whether removing stopwords and useless words such as "photo" and "image" is beneficial. This led to almost no difference since balancing entries reduced the effects of useless entries (each entry contributes to 0.0002% (1/500k) level of the total data points). To encourage a simplified solution, we do not intend to add more artificial filters.

## A.8 MORE DETAILS ON DISTRIBUTION ON METADATA

Extending the top-20 matches in Table 3, we further group counts of metadata entries and show 5 examples per group as in Table 13.

| Group | 5 Examples (Entry:Count) |
|---|---|
| 0-10k | ivailo:12, Kunta Kinte:201, vikernes:33, peria:50, ankoku:20 |
| 10k-20k | queer:19k, barry:10k, bandages:12k, The Police:15k, sigma:14k |
| 20k-50k | polygonal:21k, widely:28k, however:35k, toppers:25k, executives:21k |
| 50k-100k | planted:52k, olive oil:58k, yours:63k, packages:82k, Spokane:53k |
| 100k-500k | compact:133k, vertical:222k, underwear:111k, powder:323k, weekly:130k |
| 500k-1M | Tokyo:713k, Lead:620k, Diagram:809k, Dye:858k, unnamed:512k |
| 1M-50M | see:1.4M, Door:3.2M, News:2.3M, sea:1.1M, street:1M |
| 50M-130M | with:67M, and:100M, to:61M, in:107M, of:121M |

Table 13: Distribution of metadata entries with counts, similar as head shown in Table 3.

## A.9 RANDOMNESS OF ALGORITHM 1

We further study the randomness of algorithm 1 by running it 3 times for Pool 1(400M) and Pool 2(2.5B). This ended with a standard deviation of 4035 examples for Pool 1 and 2104 examples for Pool 2, respectively.

## A.10 QUALITATIVE DATA EXAMPLES

In Table 14, we illustrate data before/after sub-string matching and balancing. We also highlight class labels from ImageNet in the table. We mark a matched entry with a bigger font size indicating higher probability of sampling that entry. Intuitively, sub-string matching removes low quality text and balancing favors longer text with long-tail entities to improve data quality. In Table 15, we show more examples of matched text that include ImageNet tail entries.

| Text | Substr. | Bal. | IN Head | IN Tail |
|---|---|---|---|---|
| control_14ct | ✗ | ✗ | ✗ | ✗ |
| cudd2008 | ✗ | ✗ | ✗ | ✗ |
| product-img | ✗ | ✗ | ✗ | ✗ |
| Skirmisher-Main-440x412 | ✗ | ✗ | ✗ | ✗ |
| A4omote | ✗ | ✗ | ✗ | ✗ |
| How-to-Find-the-Finest-Electric-Car-Companies | ✗ | ✗ | ✗ | ✗ |
| hanukkah-party-facebook-event-cover-template | ✗ | ✗ | ✗ | ✗ |
| johnny_cash_chili_dog (2) | ✗ | ✗ | ✗ | ✗ |
| 8533 GOLDEN RIDGE COURT | ✗ | ✗ | ✗ | ✗ |
| How to build a stone patio on your own | ✓ | ✗ | ✓ | ✗ |
| battery plate | ✓ | ✗ | ✓ | ✗ |
| barn wedding | ✓ | ✗ | ✓ | ✗ |
| Picture sand , machine , Concept , jeep , the concept , the front , Slim , Wrangler , Jeep | ✓ | ✗ | ✓ | ✗ |
| desk | ✓ | ✗ | ✓ | ✗ |
| Adult T-shirt | ✓ | ✗ | ✓ | ✗ |
| Imix m10 stage moniter | ✓ | ✗ | ✓ | ✗ |
| google wallet md3 | ✓ | ✗ | ✓ | ✗ |
| Distressed beach bar sign - Pearly's Oyster Bar | ✓ | ✗ | ✓ | ✗ |
| Why the Kilimanjaro Trek should be top of your bucket list | ✓ | ✗ | ✓ | ✗ |
| J70 desk model | ✓ | ✗ | ✓ | ✗ |
| Whitby castle | ✓ | ✗ | ✓ | ✗ |
| Inside the castle | ✓ | ✗ | ✓ | ✗ |
| Hama Hama Oyster Saloon \| restaurant \| 35846 US-101 , Lilliwaup , WA 98555 , USA \| 3608775811 OR +1 360-877-5811 | ✓ | ✗ | ✓ | ✗ |
| beach | ✓ | ✗ | ✓ | ✗ |
| Caramelized onions , sauteed red bell peppers and zucchini combined create a winning egg frittata breakfast dish . | ✓ | ✓ | ✓ | ✗ |
| Vector layered paper cut craft style music composition of saxophone guitar trumpet violin music instruments , notes on abstract color background . Jazz concert festival party poster banner card template | ✓ | ✓ | ✓ | ✗ |
| Nautilus hot tub | ✓ | ✓ | ✓ | ✗ |
| night binoculars for hunting | ✓ | ✓ | ✓ | ✗ |
| 2017 cat eyes women's sunglasses for women vintage sun glasses round women sun glasses oculos oculos de sol feminino | ✓ | ✓ | ✓ | ✗ |

Table 14: Examples categorized by whether passing sub-string matching and balancing: Words in violet color are in metadata and their font size indicates probability of being sampled, ranging from 13pt that has probability close to 0, to 22pt that has probability 1. ImageNet labels in head entries are cyan.

| Text | Substr. | Bal. | IN Head | IN Tail |
|---|---|---|---|---|
| Antique German sterling silver 800 cup julep goblet, 1 | ✓ | ✓ | ✗ | ✓ |
| A journey to the East Sea by high-speed train : New KTX line makes Gangneung's wonders all the more accessible | ✓ | ✓ | ✗ | ✓ |
| little arrow design co watercolor rainbow blush shower curtain and mat | ✓ | ✓ | ✗ | ✓ |
| photo of antique silver top break revolver | ✓ | ✓ | ✗ | ✓ |
| trombone model | ✓ | ✓ | ✗ | ✓ |
| Staffordshire Bull Terrier | ✓ | ✓ | ✗ | ✓ |
| KI Plantation Timbers fire truck on the night of January 3, 2020. Photography : Tim Wilson | ✓ | ✓ | ✗ | ✓ |
| water buffalo bath | ✓ | ✓ | ✗ | ✓ |
| Basset Hound pup with Lionhead x Lop rabbit | ✓ | ✓ | ✗ | ✓ |
| Single serving of peach trifle | ✓ | ✓ | ✗ | ✓ |
| A tarantula (hairy arachnid belonging to the Theraphosidae family of spiders) in a box, standing still. Close-up shot. | ✓ | ✓ | ✗ | ✓ |
| Modern Staffordshire Bull Terrier LED Night Light Animal Pet Dog Puppy 3D Optical illusion Lamp Home Decor Table Lamp Desk Light | ✓ | ✓ | ✗ | ✓ |
| Amazon, rocking chair, CANVA, camping chairs | ✓ | ✓ | ✗ | ✓ |
| dispalying kukri machete fitted inside scabbard | ✓ | ✓ | ✗ | ✓ |
| jacksons chameleon | ✓ | ✓ | ✗ | ✓ |
| Wall Mural - Insects pollinating blooming rapeseed crops in field | ✓ | ✓ | ✗ | ✓ |
| Scottish Terrier Phone Pocket | ✓ | ✓ | ✗ | ✓ |

Table 15: Examples passing both sub-string matching and balancing wit ImageNet tail classes: Words in violet color are in metadata and their font size indicates probability of being sampled, ranging from 13pt that has probability close to 0, to 22pt that has probability 1. ImageNet labels in tail entries are in cyan.

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
