# OpenReview forum: "Demystifying CLIP Data"
_ICLR.cc/2024/Conference — ICLR 2024 spotlight_

### Official Review · Reviewer_Smdn · 2023-10-15

**Soundness:** 3 good
**Presentation:** 2 fair
**Contribution:** 3 good
**Rating:** 8
**Confidence:** 4

**Summary:**

The authors of the paper try to replicate the curation process employed by OpenAI to train their CLIP models. To that end, they highlight important excerpts from the original CLIP paper, and they try to replicate them as close as possible. The models trained on the resulting dataset show improved performance in imagenet zero-shot accuracy.

**Strengths:**

- the paper highlights several important excerpts form OpenAI's curation technique and tries to replicate them
- the authors do an ablation study over their choices to validate whether their choices are helpful
- the authors plan to release the curation data to the community

**Weaknesses:**

- the evaluation shows improvement in some downstream tasks, but not others. it would be nice to investigate this mixed performance, as to make sure that the curation technique is extremely tailored towards imagenet
- there are other proposed approaches [1] for recreating WIT-400M, however, the authors do not compare their proposed approach with other approaches

[1] DataComp: In search of the next generation of multimodal datasets, Gadre at al., 2023

**Questions:**

- how does the curation step proposed in this paper compare with the one used for LAION-400M
- can you please explain why your curation is not overfit towards ImageNet?
- can you please compare against approaches in datacomp?

---

> ### Author Response · Authors · 2023-11-20
> **Author response to Smdn (1/2)**
>
> Please note that our data is not a reproduction of OpenAI-CLIP (we cannot reproduce because the CLIP paper only has very limited information about its data sources), instead MetaCLIP makes language-image curation transparent so it can be applied to different pools to create high quality datasets which lead to better performance than CLIP, under the exact same training and model settings.
>
> Regarding the mixed performance gains we think it’s due to the following::
> - MetaCLIP vs OpenAI CLIP: the mixed performance gains are mainly due to pool sources, but since OpenAI-CLIP’s data source is unknown, it is hard to draw a conclusion;
> - MetaCLIP vs LAION/DataComp: since the source is the same, we believe the better performance of MetaCLIP has better coverage of downstream tasks with metadata balancing that favors more tasks by sampling data for long-tailed distribution, whereas LAION/DataComp has no balancing and favors head distribution, or what the OpenAI-CLIP model is biased to (which again is unknown). We hope that with open-sourcing of our full pipeline the community can develop further datasets to study performance across a variety of tasks and further our understanding in language-image pre-training.
>
> >how does the curation step proposed in this paper compare with the one used for LAION-400M
>
> MetaCLIP uses raw metadata substring matching and balancing, whereas LAION-400M uses an OpenAI-CLIP filter and text length filters (page 3 of https://arxiv.org/pdf/2111.02114.pdf), as well as hidden filters not mentioned in the paper but in the pipeline code (e.g., filters on relative URLs in parser that can be observed in the code of LAION release).
>
> LAION’s curation is based on an OpenAI-CLIP model to filter the data, therefore it distills the OpenAI-CLIP data implicitly. For example, a model trained on CLIP’s data will directly have high accuracy/similarity on the CLIP training images; therefore, this can be used to replicate CLIP’s data, but it can have low accuracy/similarity on image/text pairs that are not in CLIP’s training set. Instead, our goal is curating data from scratch. We wish to highlight the beauty of our raw curation, without (model-based) filters, to maximally preserve signal and control biases.
>
> >can you please explain why your curation is not overfit towards ImageNet?
>
> We didn’t do anything particular for ImageNet besides following OpenAI-CLIP, which uses all WordNet entries that includes ImageNet labels. The metadata contains 500k entries and ImageNet is only 1k of them; therefore, the metadata covers a large, general pool of concepts. We also de-duplicate our data against ImageNet.
>
> We want to note that this approach is a much weaker bias to ImageNet than the “Image-based filtering” used in DataComp, which, as mentioned above, searches the nearest neighbors for every ImageNet training example, and keeps examples belonging to these groups. For example, for their ViT-L experiments, the ImageNet “Image-based filtering” increases ImageNet accuracy from 76.4% to 79.2% (see Tab. 3 of the [DataComp paper](https://arxiv.org/pdf/2304.14108.pdf)).
>
> Finally, note that we report strong results on 26 benchmarks in Table 4, and on 38 benchmarks in Table 8; suggesting that our approach does not overfit to ImageNet but is generally working on multiple datasets. We however, do agree that the data distribution matters for downstream performance and performed an ablation in Table 11, where we show the number of classes that have matches in the metadata. For example, for ImageNet 703 of the 1000 classes are present in the metadata, and more general we can see that most datasets have text string matches in the very general and large (500k) set of metadata.

---

> > ### Author Response · Authors · 2023-11-20
> > **Author response to Smdn (2/2)**
> >
> > >can you please compare against approaches in datacomp?
> >
> > Please note that DataComp is a concurrent work, but we have used their data pool in the appendix of our submission and show that MetaCLIP can generalize to it. Our experiments with DataComp’s pool are shown in Tab. 4 and Fig. 4 of the appendix.
> >
> > We further compare the benchmark used in DataComp in Table 8. These are all apple-to-apple comparisons, using the same hyperparameters and models and therefore directly measure data quality.
> >
> > A direct comparison to the DataComp filtering is not possible for the following reasons:
> >
> > (1) DataComp uses different hyperparameters than OpenAI-CLIP.
> >
> > (2) DataComp curation is based on two OpenAI-CLIP filters (B and L sizes), therefore it distills the OpenAI-CLIP data implicitly (or performs data pruning).
> >
> > (3) DataComp performs “Image-based filtering” that requires clustering into 100k groups and finding the nearest neighbor group for every ImageNet training example, and keeping examples belonging to these groups.
> >
> > Having access to CLIP models and ImageNet training data is a non-trivial advantage and it has significant gains as ablated in Tab. 3 of the [DataComp paper](https://arxiv.org/pdf/2304.14108.pdf).
> > Nevertheless, please note that our MetaCLIP system, which does not need external models or data, provides similar performance on the DataComp benchmark, without any bells and whistles. For example we report 65.6% average accuracy with ViT-L in Table 8, while the best reported ViT-L result in Tab. 3 of the [DataComp paper](https://arxiv.org/pdf/2304.14108.pdf) is 66.3%, which is 65.0% without their ImageNet filter and 63.6 without their OpenAI-CLIP ViT-L filter and 62.1% without the OpenAI-CLIP ViT-B filter. This shows that our approach is competitive and much more accurate without reliance on external models.
> >
> > We thank the reviewer for their insightful review and appreciate positive comments about the paper. Please let us know if anything else is unclear and we will try to answer.

---

> > > ### Comment · Reviewer_Smdn · 2023-11-22
> > >
> > > Thank you for the clarification. My impression of the paper remains positive and I will raise my score. I suggest editing the paper to include the above clarifying points, especially the comparison with DataComp.

---

### Official Review · Reviewer_f2FV · 2023-10-21

**Soundness:** 3 good
**Presentation:** 3 good
**Contribution:** 2 fair
**Rating:** 6
**Confidence:** 5

**Summary:**

This paper attempts to uncover the data curation process of the CLIP paper by Radford etc. The curation process is composed of several steps: 1) 500k entry construction 2) text2entry matching 3) entry2text indexing 4) entry balancing. Applying this pipeline to the CommonCrawl dataset, the resulting dataset achieves even better performance CLIP.

**Strengths:**

1. After almost three years the CLIP paper came out, it is great to see efforts following and investigating the data curation pipeline in the CLIP paper, whose proposed data diversification (balancing) was ignored by the other works, such as LAION. The paper would be interesting to the researchers working on data curation, and contrastive pre-training, too.
2. The experimental results are impressive and set the new state of the art.

**Weaknesses:**

1. The contribution of the paper is limited. The main contribution is the entry balancing used by CLIP. This balancing operation actually plays a similar role to the deduplication used in [1] and [2], where its effectiveness has been proven.
2. In Sec 3.4, when sub-sampling image-text pairs for each entry, in addition to the information density based rule, it is worth trying some model-based rules, e.g., image-text matching based rules. Although the paper mainly aims to reproduce the CLIP paper's results, it would lead to more contributions to go beyond CLIP reproduction and curate higher-quality datasets.

[1] Abbas, Amro, et al. "SemDeDup: Data-efficient learning at web-scale through semantic deduplication." arXiv preprint arXiv:2303.09540 (2023).
[2] Yu, Haichao, et al. "The Devil is in the Details: A Deep Dive into the Rabbit Hole of Data Filtering." arXiv preprint arXiv:2309.15954 (2023).

**Questions:**

1. How many runs are conducted to get the quantitative experimental results in the paper (e.g., Table 4 and 5)? Are the standard deviations sufficiently smaller than the differences between different settings?
2. Any analysis on where the improvement over CLIP dataset is from in Table 4? Different raw data sources?

---

> ### Author Response · Authors · 2023-11-20
> **Author response to f2FV**
>
> >The contribution of the paper is limited. The main contribution is the entry balancing used by CLIP. This balancing operation actually plays a similar role to the deduplication used in [1] and [2], where its effectiveness has been proven.
>
> The contribution of this paper is the full MetaCLIP curation pipeline (that creates a structured dataset based on large-scale metadata), which is novel to the community and allows to have full transparency and control for image-language pretraining. This is developed on top of the limited information present in the CLIP paper (note that the original CLIP paper does not describe the data curation pipeline and only contains three paragraphs about it, in the 48-page (Radford et al, 2021) paper). Besides that, our curation is scalable to the whole Internet and outperforms CLIP’s data by a significant margin.
>
> From the balancing perspective, [1, 2] use trained CLIP embeddings as features for K-means clustering and perform duplicate removal via excluding examples too close to cluster centers. This post-hoc filtering may carry CLIP’s unknown biases (e.g., see discussion in [LAION-5B Appendix G.2](https://arxiv.org/pdf/2210.08402.pdf)) and MetaCLIP has balancing via matching unstructured, noisy text. The main difference between MetaCLIP and [1, 2] is that these works are filtering methods that filter data using heuristics and the CLIP model. Therefore, they rely implicitly on the unknown CLIP data. Our method instead directly creates the data in a transparent Algorithm, without reliance on filters or external models.
>
>
> >it is worth trying some model-based rules, e.g., image-text matching based rules. Although the paper mainly aims to reproduce the CLIP paper's results, it would lead to more contributions to go beyond CLIP reproduction and curate higher-quality datasets.
>
> These are great ideas to further increase data quality; however, our goal is to make CLIP curation transparent: using OpenAI-CLIP adds a black-box filter back and thus conflicts with our initial goal. For example, a model trained on CLIP’s data will directly have high accuracy/similarity on the CLIP training images; therefore, this can be used to replicate CLIP’s data, but it can have low accuracy/similarity on image/text pairs that are not in OpenAI-CLIP’s training set.
> Instead, our goal is curating data from scratch. We wish to highlight the beauty of our raw curation, without (model-based) filters, to maximally preserve signal and control biases.
>
> >How many runs are conducted to get the quantitative experimental results in the paper (e.g., Table 4 and 5)? Are the standard deviations sufficiently smaller than the differences between different settings?
>
> Yes, significantly smaller. The standard deviation of multi seed training is +-0.1% acc for ImageNet on ViT-B/32. We added this information to the paper.
>
> >Any analysis on where the improvement over CLIP dataset is from in Table 4? Different raw data sources?
>
> Unfortunately, we have no access to OpenAI-CLIP's training data for direct comparison. Yes, we also think it is because of different raw data sources: Our accuracy over the different downstream datasets appears closer to LAION, since it is based on the same CommonCrawl source. We have no information about OpenAI-CLIP’s source, but would be happy to perform more analysis for the final paper.
>
> We thank the reviewer for their insightful review and suggestions for improvement. Please let us know if anything else is unclear and we will try to answer.

---

> > ### Comment · Reviewer_f2FV · 2023-11-22
> >
> > Thanks for the replies. Overall, this paper is a great step towards reproducible vision-language data curation. Increased my rating.

---

### Official Review · Reviewer_33mj · 2023-11-01

**Soundness:** 3 good
**Presentation:** 4 excellent
**Contribution:** 4 excellent
**Rating:** 8
**Confidence:** 4

**Summary:**

This paper aims at reproducing the data collection and construction process of the original CLIP, which is still unknown to the reserach community, despite of the brief introduction in the CLIP paper, and some followup public datasets like LAION or DataComp. Specifically, it first collects some high-quality metadata based on Wikipedia and WordNet, uses them to retrive text entry from snapshots of CommonCrawl, and employs a balancing strategy to build a more diverse and balanced data distribution. The CLIP model trained on the resulting dataset shows comparable or superior performance compared to CLIP or OpenCLIP.

**Strengths:**

1. The effort to reproduce the exact data construction procedure of the original CLIP paper is well motivated and appreciated. As the authors pointed out, the later datasets like LAION or DataComp, all adopt trained CLIP model during data collection. How to build a high-quality and diverse image-text dataset from scratch, like WIT400M, is still a mystery to the community.
2. Given that CLIP is such a important foundation model that connects image and text, the data crafting pipeline and the resulting dataset are of clear importantce to the community. Also, the lessons learnt in this work could potentially benefit future efforts to construct high-quality open datasets in other areas as well.
3. The writing is good. The overall logistics follow a reasonable thread of investigation.
4. The performance of the model trained on MetaCLIP often show superior performance compared to LAION or WIT.

**Weaknesses:**

1. I find that the authors use the average accuracy across multiple datasets as a major performance metric throughout the paper. This is examplified by table 4/5 in the main texts, and also some tables in the appendix. This does not make sense to me. Those datasets come with different number of classes and number of samples. For instance, averaging the accuracy of a dataset of 10 classes (e.g. EuroSAT), and a dataset of 102 classes (e.g. Flowers), is unreasonble, because misclassifying all samples in one class corresponds to 10% accuracy in the former dataset, but corresponds to ~1% accuracy in the former dataset. I know that some other works, like DataComp, also keep this practice. But I feel a more reasonable approach is to compare the performance on individual datasets, and how many datasets on which the MetaCLIP model prevails. As such, the authors should list results on individual datasets, not report some average accuracies.

2. Following previous one, I find results in table 8 in appendix tend to be more comprehensive, as the model is also evaluated on ImageNet distribution shift datasets and retrieval datasets. The organization of this paper can be enhanced if the table 4/5 in the main text also report results on those dataset.

3. Some data filtering details could be further detailed. For example, the last sentence in page 4 (sorry but this submission does not have a line number) talks a bit about some *noise reduction... (e.g., regular expression on dates, ids etc.)*. Also, figure 5 in the appendix shows some *URL & Text Dedup* and *NSFW Image Dedup*, but does not describe them in details. The authors are encouraged to list more detailed descriptions on those noise reduction and deduplication processes, and also explain their effects on the resulting dataset.

**Questions:**

1. How does different choices of metadata affect the final dataset? I know the authors follow the procedure listed in the original CLIP paper, but I feel like the decision to use Wikipedia and WordNet can be futher explained by 1) either a comprehensive distribution probing or visualization of the metadata used in the paper (right now only top-20 entried are showed, and they seem very general and vague wordings); 2) remove some metadata sources (like no wordnet synset), or tune the threshold hyper-parameters like PMI or view frequency.

2. Could in data construction process inspire data creating efforts in other areas? The authors are encouraged to add some related discussion, possibly in a *broader impact* section.

3. In section 4, it is mentioned multiple times that using a smaller *t* can *increases tail metadata matches*, like the last sentence in page 7. But my understanding is that a smaller *t* only leads less matches of head metadata, while all the matches of tail metadata are kept. Am I understanding it wrong?

4. Since randomness is introduced in the balancing stage, I wonder how much impact does it have with different random seeds?

5. Will the authors released detailed code (not just some demo) to reproduce their whole pipeline of data curation from scratch? For this work, I think it is not the resulting dataset MetaCLIP that is most valuable. How the authors arrive at the final dataset from some wiki pages and wornet synets to the final collection of diverse and high-quality image-text pairs really matters, and could make a big difference and push the community forward once open sourced.

---

> ### Author Response · Authors · 2023-11-20
> **Author response to 33mj (1/2)**
>
> >I find that the authors use the average accuracy across multiple datasets as a major performance metric throughout the paper.
>
> We agree with the drawback of reporting average accuracy, but we are following this as it is standard practice in the community. We created a version of Table 8 that reports on all individual datasets and included it in the updated paper.
>
> >Some data filtering details could be further detailed. For example, the last sentence in page 4 (sorry but this submission does not have a line number) talks a bit about some noise reduction... (e.g., regular expression on dates, ids etc.).
>
> In this part of the text we only state that sub-string matching to metadata automatically removes noisy captions (e.g. a caption just containing a date string), without using complicated hand-crafted filters (e.g. a regular expression that removes such cases).
>
> >Also, figure 5 in the appendix shows some URL & Text Dedup and NSFW Image Dedup, but does not describe them in details.
>
> We will provide more details on our pipeline in the paper and will add the following to appendix A.2.:
> - Our HTML Parser is applied to all WAT files of CommonCrawl by keeping all <img> tags with both relative and absolute URLs and one or multiple text keys.
> - Language-Identification: we use an internal language identification system that can detect 191 languages with different dialects and keep all texts that are tagged as English (or its dialects).
> - URL / Text deduplication: We use 24bit sha224 hashing to encode an URL and into tables to deduplicate columns with the same hash, avoiding downloading the same image URL multiple times. URLs with illegal domains are removed; if there are multiple texts associated with the same image, these are further deduplicated; texts with NSFW keywords are removed;
> - NSFW filter: We use an internal system that can classify inappropriate content in images into 96 types of dangerous content and discard such image/text pairs.
> - Image Deduplication: Images are further deduplicated by 64-bit PCA hash, derived from a similarity search model's feature embeddings with PCA reduction to 64 dimensions and sign quantization.
>
> >1) either a comprehensive distribution probing or visualization of the metadata used in the paper (right now only top-20 entried are showed, and they seem very general and vague wordings);
>
> Here we further group counts of entries and show 5 examples per group as below:
>
> | Count Group | 5 Examples (Entry:Count) |
> |:-----------|:-----------------------------------------------------:|
> 0-10k | ivailo:12, Kunta Kinte:201, vikernes:33, peria:50, ankoku:20 |
> 10k-20k | queer:19k, barry:10k, bandages:12k, The Police:15k, sigma:14k |
> 20k-50k | polygonal:21k, widely:28k, however:35k, toppers:25k, executives:21k |
> 50k-100k | planted:52k, olive oil:58k, yours:63k, packages:82k, Spokane:53k |
> 100k-500k | compact:133k, vertical:222k, underwear:111k, powder:323k, weekly:130k |
> 500k-1M | Tokyo:713k, Lead:620k, Diagram:809k, Dye:858k, unnamed:512k |
> 1M-50M | see:1.4M, Door:3.2M, News:2.3M, sea:1.1M, street:1M |
> 50M-130M | with:67M, and:100M, to:61M, in:107M, of:121M |
>
> This Table is now also in the appendix of the updated paper (Table 13).
>
> The counts exhibit a long-tailed distribution which is shown in the plot of the cumulative sum of distribution in Fig 2/Fig. 3. We release the full distribution with our code.
>
> >2) remove some metadata sources (like no wordnet synset), or tune the threshold hyper-parameters like PMI or view frequency.
>
> This is a great suggestion; however, note that ablating metadata is challenging, as this is the first step of the whole data pipeline and therefore it is difficult to backtrack quantitative signals from the downstream tasks to this step.
>
> We conducted the suggested experiment by removing wordnet from the metadata (this results in removal of 45786 (out of 86554 wordnet entries). This removes 9M training data from the MetaCLIP-400M dataset (because wikipedia sources can cover certain wordnet entries, or one text with wikipedia matched entry can carry a wordnet entry without match). We are training on this set and will report the result later. After 6B/12.8B samples seen (iterations) the accuracy is 55.3% vs 55.6% with full 400M data scale on ImageNet.

---

> > ### Author Response · Authors · 2023-11-20
> > **Author response to 33mj (2/2)**
> >
> > >Could in data construction process inspire data creating efforts in other areas? The authors are encouraged to add some related discussion, possibly in a broader impact section.
> >
> > MetaCLIP explores a scalable and high-quality data curation strategy for CLIP training. We agree that Algorithm 1, or some variant thereof, has the potential for seamless integration into various fields. Beyond the image-text scenario examined in this work, we foresee that MetaCLIP can be applied to  web-scale video-text and text-only pretraining. Furthermore, this approach is envisioned to be applicable to both discriminative and generative pre-training objectives, underscoring the increasing importance of large-scale high-quality data in the training of modern large models. We appreciate your insightful suggestion and will include the content in the broader impact section.
> >
> > >In section 4, it is mentioned multiple times that using a smaller t can increases tail metadata matches, like the last sentence in page 7. But my understanding is that a smaller t only leads less matches of head metadata, while all the matches of tail metadata are kept. Am I understanding it wrong?
> >
> > Your understanding is correct; The increased tail metadata matching is the result of using a larger pool (e.g, Pool 2, since with more data, a larger pool is more likely to fill tail matching, as mentioned in the last sentence in page 7).
> >
> > >Since randomness is introduced in the balancing stage, I wonder how much impact does it have with different random seeds?
> >
> > Good question, for 400M data (pool 1), 3 different seeds result in +-4035 examples standard deviation and +- 2105 examples for the 2.5B data. We are training on these 3 different train sets and will report results later in the final version.
> >
> > >Will the authors released detailed code (not just some demo) to reproduce their whole pipeline of data curation from scratch? For this work, I think it is not the resulting dataset MetaCLIP that is most valuable. How the authors arrive at the final dataset from some wiki pages and wornet synets to the final collection of diverse and high-quality image-text pairs really matters, and could make a big difference and push the community forward once open sourced.
> >
> > We fully agree with this. We are actively working on releasing the full pipeline.
> >
> > We would like to thank the reviewer for their time and detailed review. Please let us know if anything else is unclear and we will try to answer.

---

> > > ### Comment · Reviewer_33mj · 2023-11-20
> > >
> > > Thanks for the reply! I recognize the effort the authors need to take for preparing the response. After reading the rebuttal, I feel that this work is worthwile a publication at ICLR.
> > >
> > > My remaining concern is that if the authors will indeed release the whole implementation from scratch. As mentioned by other reviewers, the method in this work has already been proposed by CLIP. But OpenAI refuses to share how they create the WIT data, and that makes this work, which focuses on how to craft a high-quality image-text dataset from scratch, valuable to the community. I see in the authors' rebuttal that many times they use some internal systems or private models for filtering and deduplication. That is no good to the open-source community. The authors should try their best to enable outsiders who have no access to such system or model to be able to reproduce their resulting datasets. Without such code releasing, the value of this work is greatly compromised.
> > >
> > > At this point, my original rating continues to apply.

---

> > > > ### Author Response · Authors · 2023-11-21
> > > > **full open-source pipeline**
> > > >
> > > > Thanks for your response. We are trying our best to make the full pipeline available; however, the internal tooling is in place to meet minimal legal requirements and we cannot open source that. We are confident the internal components are not implicating performance in a significant manner and that they can be replaced by existing open-source counterparts (we do expect similar or better performance with these), specifically:
> > > > - Our open-sourced parts can be integrated with [cc2dataset](https://github.com/rom1504/cc2dataset) to perform URL deduplication; text deduplication associated with one URL is a simple python set([<list of captions belonging to the same URL>]).
> > > > - Image downloading can be replaced by [img2dataset](https://github.com/rom1504/img2dataset).
> > > > - NSFW filtering and image deduplication can be replaced by [Datacomp’s image dedup](https://github.com/mlfoundations/dataset2metadata).
> > > >
> > > > We are actively working on integrating our code release with 3rd party solutions. We already released our main pipeline (but cannot share for anonymity), including our parser, language-identification, curation code, data loader and training code. Please let us know if you have any more questions. Thank you.

---

### Official Review · Reviewer_7LK5 · 2023-11-01

**Soundness:** 3 good
**Presentation:** 3 good
**Contribution:** 3 good
**Rating:** 5
**Confidence:** 3

**Summary:**

The paper proposes a data curation pipeline for CLIP training. The paper evaluates the model performance resulting from different data curation pipelines, and shows that the proposed pipeline outperforms closed-source CLIP, as well as Open CLIP. Moreover, an interesting finding from the paper is that balancing the metadata distribution is the key to improving performance.

**Strengths:**

- The paper tackles an important problem in the community regarding the opacity of data curation processes of foundation models. Moreover, it promises to open-source part of the efforts, including data curation code and training data distribution.

- The paper is also strong in terms of empirical evaluation, including various data sizes, and different implementations of the balancing steps. Huge resources are devoted to the evaluation part.

**Weaknesses:**

- The main weakness lies in the technical novelty. The paper is a commendable effort to reproduce the data curation pipeline that has already been described in the original CLIP paper (Radford et al, 2021), and report the findings for reproducing an existing data curation technique. The paper's novelty can be greatly enhanced by exploring some new technical components beyond what's already described in Radford et al 2021.

**Questions:**

1. Page 3"this post-hoc data pruning approach has limited utility, as the computational resources saved have already been expended during the initial training of the model." The reviewer has some concern regarding this comment. While posthoc data pruning still requires training an initial model, the pruned dataset can also allow efficient model improvement, which is usually an iterative process involving experimentation with different model hyperparameters.

2. Is there a specificreason to omit some other data curation baselines proposed in the DataComp paper?

3. What is the meta data distribution for OpenCLIP?

---

> ### Author Response · Authors · 2023-11-20
> **Author response to 7LK5**
>
> >The paper is a commendable effort to reproduce the data curation pipeline that has already been described in the original CLIP paper (Radford et al, 2021), and report the findings for reproducing an existing data curation technique.
>
> We would like to note that the original CLIP paper does not describe the data curation pipeline. We are following (and quoting in the submission) *all* information about CLIP’s data curation, which is extremely limited (three paragraphs in the 48-page (Radford et al, 2021) paper). Our full MetaCLIP curation pipeline presented in the paper is developed on top of this limited information and novel to the community. This has not existed before, instead the community has been using CLIP models to reproduce its data.
>
> Besides this new and scalable curation approach (e.g. Alg. 1 runs at Internet scale of 300+B pairs and performs online curation in a data loader in Table 6) we believe that the scope of novelty is broad and is beyond technical algorithms. It can also include extending existing paradigms to new problems, designing new experiments/ablations, drawing new observations/insights, and verifying expected hypotheses in new problems or tasks. From the reviewer comments, we have seen that many of these values of our work are recognized. We thank the reviewer for seeing value in our work from that perspective.
>
> >Page 3"this post-hoc data pruning approach has limited utility, as the computational resources saved have already been expended during the initial training of the model." The reviewer has some concern regarding this comment. While posthoc data pruning still requires training an initial model, the pruned dataset can also allow efficient model improvement, which is usually an iterative process involving experimentation with different model hyperparameters.
>
> We thank the reviewer for pointing this out and rephrased accordingly. Our main point here was that costs associated with training a model for filtering (e.g. a CLIP model) should also be taken into account. We agree on the value of data pruning, especially with open models and known training data.
>
> >Is there a specific reason to omit some other data curation baselines proposed in the DataComp paper?
>
> Please note that DataComp is a concurrent work, but we used their data pool and show that MetaCLIP can generalize to it in our appendix. Our experiments with DataComp’s pool are shown in Tab. 4 and Fig. 4 of the Appendix.
>
> We further compare the benchmark used in DataComp in Table 8. These are all apple-to-apple comparisons, using the same hyperparameters and models and therefore directly measure data quality.
>
> A direct comparison to the DataComp filtering is not possible for the following reasons: (1) DataComp uses different hyperparameters than OpenAI-CLIP. (2) DataComp curation is based on two OpenAI-CLIP filters (B and L sizes), therefore it distills the OpenAI-CLIP data implicitly. (3) DataComp performs “Image-based filtering” that requires clustering into 100k groups and finding the nearest neighbor group for every ImageNet training example, and keeping examples belonging to these groups. Having access to CLIP models and ImageNet training data is a non-trivial advantage and it has significant gains as ablated in Tab. 3 of the [[DataComp paper](https://arxiv.org/pdf/2304.14108.pdf)](https://arxiv.org/pdf/2304.14108.pdf). For example, for their ViT-L experiments, the “Image-based filtering” increases ImageNet accuracy from 76.4% to 79.2%. Nevertheless, please note that our MetaCLIP system, which does not need external models or data, provides similar performance on the DataComp benchmark, without any bells and whistles.
>
> >What is the meta data distribution for OpenCLIP?
>
> OpenCLIP does not have a directly accessible distribution as it  uses the OpenAI-CLIP model to filter data for creating the LAION dataset. Therefore implicitly it carries the OpenAI-CLIP distribution, which is unknown because OpenAI did not release it.
>
> We would like to thank the reviewer for their insightful review. Please let us know if anything else is unclear and we will try to answer.

---

### Author Response · Authors · 2023-11-20
**Author response summary**

We thank all reviewers for their thorough reviews and detailed suggestions. We especially appreciated positive comments, e.g., *“tackles an important problem”, “outperforms closed-source CLIP”, “ strong in terms of empirical evaluation”, “of clear importance to the community”, “benefit future efforts to construct high-quality open datasets in other areas”, “superior performance compared to LAION or WIT”, “interesting to the researchers working on data curation, and contrastive pre-training, too”, “experimental results are impressive and set the new state of the art”*.

We also thank the reviewers for their constructive criticism. We have made changes to the draft based on the suggestions (we highlighted all changes in the updated pdf in *blue*) and have collected our individual responses below.

---

### Meta-Review · Area_Chair_krF4 · 2023-12-07

**Metareview:**

**Summary**

This submission focus on reproducing the data pipeline for CLIP training. Common belief is that training data has become the most important factor (compared to modeling and training objective) for foundation models. However, an extremely popular model CLIP's training pipeline remains unknown to date (more than 2 years since the first release), with merely brief, high-level textual description from its paper. Existing approaches proposed open source version, but requires CLIP for data filtering. On the other hand, this work build the pipeline from scratch, thus uncovering the mystery of how to create a good dataset for vision-language contrastive pre-training. More specifically, high quality metadata is extracted from Wikipedia and WordNet, and then used to retrieve data from the CommonCrawl. A strategy to enforce diverse and balanced data is performed. Empirically, the resulting model, named MetaCLIP, achieves stronger performance than the original closed source CLIP and OpenCLIP.

**Strengths**
- This submission tackles an increasingly important but under-studied problem -- how to create dataset for foundation model training.
- The authors has promised to open-source as much as possible. The community will benefit a lot from this.
- Thorough empirical evaluation is conducted. Fair comparison has been made to ensure that data is the only factor for performance difference. The results are valuable because they take a lot of computing resources.
- Prior works such as LAION and DataComp relied on CLIP for the data, which is kind of problematic since one first need data to train CLIP. MetaCLIP did a great job not relying on the original CLIP, thus lessons learned in this work could be beneficial for future work's data collection.
- In general, this is a well written paper


**Weaknesses**
- Some reviewers complained about the limited novelty, since it's a reproduction of CLIP. As an AC, I don't see novelty in high priority. In fact, I think *contribution* is the more important criteria. This submission did tell us something we did not know (how to create CLIP training data), and CLIP-like foundation model has been proven useful in many areas. The contribution of this work is definitely sufficient to outweigh the abstractive, subjective novelty measure.
- Some implementation details and comparison/discussion w.r.t previous work is not very clear in the initial version. I encourage the author refine the manuscript to include these details provided in rebuttal.

**Justification For Why Not Higher Score:**

Although I think this work achieves great contribution to the community, it is still studying a 2+ year old model (CLIP). What the community cares more about now is probably how to create data for GPT-V(ision) or DALL-E 3. Therefore, I don't think this submission makes an oral paper.

**Justification For Why Not Lower Score:**

I think this paper worth a place in spotlight papers because the study of data has always been under-valued in academia / open-source research. This submission did a great job to uncover those data secrets from proprietary models. One might argue that this data is not completely new due to the existence of LAION / OpenCLIP. Still, I think the bottom line for this submission is a clear accept.

---

### Decision · Program_Chairs · 2024-01-16

Accept (spotlight)